# LLaMA-Adapter:
# Efficient Fine-tuning of Large Language Models with Zero-initialized Attention

**Renrui Zhang**[*1,2], **Jiaming Han**[*1,2], **Chris Liu**[*1], **Aojun Zhou**[2], **Pan Lu**[3]
**Yu Qiao**[†1], **Hongsheng Li**[†2,4], **Peng Gao**[†‡*1]

[1]Shanghai Artificial Intelligence Laboratory  [2]CUHK MMLab
[3]University of California, Los Angeles  [4]CPII of InnoHK

```
{zhangrenrui, hanjiaming, gaopeng, qiaoyu}@pjlab.org.cn
hsli@ee.cuhk.edu.hk
```

## Abstract

With the rising tide of large language models (LLMs), there has been a growing interest in developing general-purpose instruction-following models, e.g., Chat-GPT. To this end, we present **LLaMA-Adapter**, a lightweight adaption method for efficient instruction tuning of LLaMA. Using 52K self-instruct demonstrations, LLaMA-Adapter only introduces **1.2M** learnable parameters upon the frozen LLaMA 7B model, and costs less than **one hour** for fine-tuning. Specifically, a zero-initialized attention mechanism is proposed. It adopts a learnable zero gating to adaptively inject the instructional cues into LLaMA within self-attention layers, contributing to a stable training process and superior final performance. In this way, LLaMA-Adapter can generate high-quality responses to diverse language instructions, comparable to Alpaca with fully fine-tuned 7B parameters. Besides language commands, by incorporating an image encoder, our approach can be simply extended to a **Multi-modal LLM** for image-conditioned instruction following, which achieves superior multi-modal reasoning capacity on several popular benchmarks (MME, MMBench, LVLM-eHub). Furthermore, we also verify the proposed zero-initialized attention mechanism for fine-tuning other pre-trained models (ViT, RoBERTa, CLIP) on traditional vision and language tasks, demonstrating the effectiveness and generalizability of our approach. Code and models are released at https://github.com/OpenGVLab/LLaMA-Adapter.

## 1 Introduction

Large Language Models (LLMs) (Dai et al., 2019; Radford et al., 2019; Zhang et al., 2022; Raffel et al., 2020; Devlin et al., 2018) have stimulated widespread attention in both academia and industry. Driven by massive corpora and advanced hardware, LLMs exhibit remarkable understanding and generative ability, propelling language tasks to a higher level. Recently, significant progress has been made on instruction-following models, e.g., ChatGPT (OpenAI, 2023a) and GPT-4 (OpenAI, 2023b), which follow language instructions and generate contextual responses. However, the further prevalence of instruction models is largely impeded by the closed-source restriction and high development costs.

To alleviate this, Stanford Alpaca (Taori et al., 2023) proposes to fine-tune an open-source LLM, i.e., LLaMA (Touvron et al., 2023) into an instruction-following model, which is affordable and replicable. Starting from 175 human-written instruction-output pairs (Wang et al., 2022a), Alpaca leverages GPT-3.5 (Brown et al., 2020) to expand the training data to 52K in a self-instruct manner. Supervised by this, Alpaca fine-tunes the entire 7B parameters in LLaMA, producing an exceptional instruction model that performs similarly to GPT-3.5. Despite Alpaca's effectiveness, a complete fine-tuning of large-scale LLaMA is still time-consuming, computation-intensive, and cumbersome to transfer to different downstream scenarios.

---

[*] Equal contribution  [†] Corresponding author  [‡] Project leader

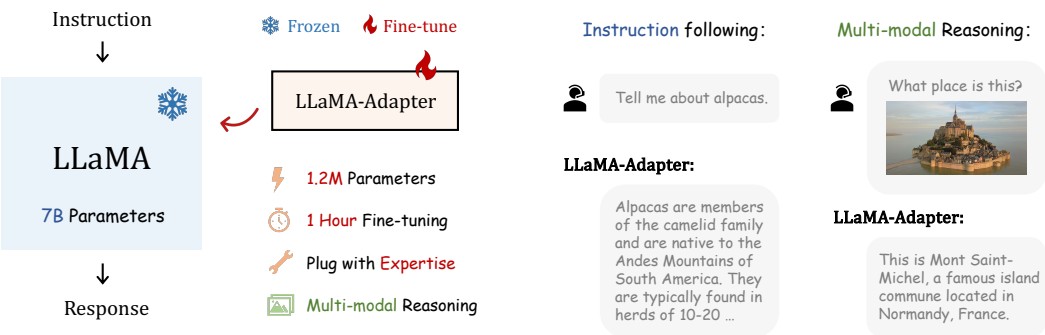

Figure 1: **Characteristics of LLaMA-Adapter.** Our lightweight adaption method efficiently fine-tunes LLaMA (Touvron et al., 2023) 7B model with only 1.2M learnable parameters within one hour, which exhibits superior instruction-following and multi-modal reasoning capacity.

In this paper, we introduce **LLaMA-Adapter**, an efficient fine-tuning method that adapts LLaMA into a well-performed instruction-following model. Trained by Alpaca's instruction-output data, our approach freezes the entire LLaMA model, and proposes a zero-initialized attention mechanism with superior resource efficiency. Specifically, in LLaMA's higher transformer layers, we append a set of learnable adaption prompts as prefixes to the word tokens. Then, to avoid the noise from randomly initialized prompts at the early training stage, we equip the frozen self-attention layers with a learnable gating factor. The gating mechanism is initialized by zeros, and controls the feature interaction between prompt and word tokens, within the process of attention calculation. Such a strategy can first preserve the original knowledge in LLaMA, and progressively inject the new instructional signals during training. This contributes to a more stable learning process and better instruction-following capacity of the final model.

Overall, our LLaMA-Adapter exhibits four main characteristics, as shown in Figure 1.

- **1.2M Parameters.** Instead of updating the full 7B parameters, we freeze the pre-trained LLaMA and only learn the zero-initialized attention mechanism with 1.2M parameters. This, however, reveals comparable instruction-following proficiency with the 7B Alpaca.

- **One-hour Fine-tuning.** Thanks to our lightweight adaption modules with zero-initialized gating, the training convergence of LLaMA-Adapter costs less than one hour on 8 A100 GPUs, which are three times faster than Alpaca.

- **Plug with Expertise.** For different scenarios, it is flexible to insert their respective adapters to endow LLaMA with different expert knowledge or new modality input. Thus, it suffices to store a 1.8M adapter within each context, other than a complete copy of the 13G LLaMA.

- **Multi-modal Reasoning.** Besides language instruction, our approach can also incorporate an image encoder via zero-initialized attention to become a multi-modal LLM. Compared to concurrent works (Liu et al., 2023b; Zhu et al., 2023), LLaMA-Adapter showcases higher tuning efficiency with competitive reasoning capacity on MME (Fu et al., 2023), MMBench (Liu et al., 2023c), and LVLM-eHub (Xu et al., 2023) benchmarks.

In addition to instruction tuning, our zero-initialized attention can be generalized to traditional vision and language tasks for parameter-efficient fine-tuning. We apply our approach to the pre-trained ViT (Dosovitskiy et al., 2020), ReBERTa (Liu et al., 2019), and CLIP (Radford et al., 2021), respectively for fine-tuning vision, language, and vision-language models. On a wide range of downstream tasks, we demonstrate the effectiveness of our proposed method for traditional tasks.

## 2    RELATED WORK

**Instruction Tuning of Language Models.**   The subfield of language models learning instruction-following capabilities aims to generate responses based on natural language commands. These methods normally enhance the pre-trained LLMs by fine-tuning them with high-quality instruction-output data pairs. Early works, such as FLAN (Wei et al., 2021), PromptSource (Bach et al.,

2022), and SUP-NATINST (Wang et al., 2022b), introduce effective instruction tuning methods and establish comprehensive evaluation benchmarks. InstructGPT (Ouyang et al., 2022) demonstrates significant improvement in the instruction-following power, but is closed-source to the community. To promote the open source of instruction models, Stanford Alpaca (Taori et al., 2023) fine-tunes all the 7B parameters of LLaMA (Touvron et al., 2023) with 52K self-instruct data. However, this full-model fine-tuning can be inefficient in both time and computation resources, limiting its transferability to downstream applications. In this paper, we propose LLaMA-Adapter to fine-tune only lightweight zero-initialized attention mechanisms on top of the frozen LLaMA, other than updating parameters of the entire model. There are several works **_concurrent_** to ours, Alpaca-LoRA (alp, 2023), Vicuna (Chiang et al., 2023), and LLaMA-GPT4 (Peng et al., 2023), which aim to improve Alpaca from different aspects. Alpaca-LoRA utilizes the existing LoRA (Hu et al., 2021) to efficiently fine-tune LLaMA, which is restricted to the original network structure and cannot be extended for image input. In contrast, our LLaMA-Adapter achieves higher training efficiency and can be simply generalized to a multi-modal LLM via zero-initialized attention. Vicuna and LLaMA-GPT4 target at constructing a more advanced instruction dataset using ChatGPT (OpenAI, 2023a) and GPT-4 (OpenAI, 2023b), instead of Alpaca's 52K data, which still adopt full fine-tuning without the potential for multi-modal instruction tuning.

**Parameter-efficient Fine-tuning.** The pre-training and fine-tuning paradigms have been proven to be highly effective in different language and vision tasks. Compared to full fine-tuning, Parameter-Efficient Fine-Tuning (PEFT) (Paul, 2022) methods freeze most parameters of pre-trained models, and aim to exhibit comparable capabilities on downstream tasks (Wang et al., 2018; Puzikov & Gurevych, 2018). Therein, prompt tuning appends a collection of trainable tokens to pre-trained large models, which are inserted either to the input embeddings (Lester et al., 2021; Liu et al., 2021b) or every intermediate layer (Li & Liang, 2021; Liu et al., 2021a). LoRA (Hu et al., 2021; Zhang et al., 2023d; Hedegaard et al., 2022) introduces trainable rank decomposition matrices into each network weights (Karimi Mahabadi et al., 2021), indicating promising fine-tuning ability on large generative models (Cuenca & Paul, 2023; alp, 2023). Adapters (Houlsby et al., 2019) insert lightweight adaption modules into each block of the transformer and have been extended across numerous domains (Gesmundo & Dean, 2022; Gao et al., 2021; Zhang et al., 2021). Different from previous efforts, we propose the LLaMA-Adapter with zero-initialized attention specially designed for instruction tuning and multi-modal reasoning of LLaMA (Touvron et al., 2023). Some existing works also adopt gating techniques in prompt tuning (Yoo et al., 2023; Goswami et al., 2023), but conduct a naive gated combination of different prompt tokens with randomly initialized factors. Instead, our gating factor learns from zero during training, and is delicately integrated into self-attention layers. Another branch of work applies zero initialization to convolutional networks (Zhao et al., 2021), text-to-image diffusion models (ControlNet (Zhang et al., 2023c)), or vision-language learning (Flamingo (Alayrac et al., 2022)). They are not PEFT methods requiring large-scale parameters, and have motivations for better network-level initialization or feature-level fusion via residual connections, very different from our interaction controlling within attention layers.

## 3 LLaMA-Adapter

In Section 3.1, we first introduce to insert learnable adaption prompts into LLaMA's (Touvron et al., 2023) transformer. Then, we present the details of zero-initialized attention mechanisms with zero gating in Section 3.2, and generalize LLaMA-Adapter for multi-modal reasoning in Section 3.3. Finally, we extend our approach for efficient fine-tuning of language and vision models in Section **??**.

### 3.1 LEARNABLE ADAPTION PROMPTS

Given a pre-trained LLaMA with an $N$-layer transformer, we first insert a set of learnable adaption prompts into its topmost $L$ layers ($L \leq N$). We denote the prompts as $\{P_l\}_{l=1}^{L}$, where $P_l \in \mathbb{R}^{K \times C}$ with $K$ denoting the prompt length for each layer, and $C$ equaling the feature dimension of LLaMA's transformer. The prompting at last $L$ layers can better tune the language representations with higher-level semantics.

Taking the $l$-th inserted layer as an example ($l \leq L$), we denote the $M$-length word tokens as $T_l \in \mathbb{R}^{M \times C}$, which represent the input instruction and the already generated response. The learnable

adaption prompt is concatenated with $T_l$ along the token dimension as prefixes, formulated as

$$[P_l; \ T_l] \ \in \mathbb{R}^{(K+M)\times C}. \tag{1}$$

In this way, the instruction knowledge learned within $P_l$, can effectively guide $T_l$ to generate the subsequent contextual response via our zero-initialized attention layers in the transformer block.

## 3.2 ZERO-INITIALIZED ATTENTION

If the adaption prompts are randomly initialized, they might bring disturbance to the word tokens at the beginning of training, which harms the fine-tuning stability and effectiveness. Considering this, we modify the vanilla self-attention at the last $L$ layers to be zero-initialized variants, as shown in Figure 2. Suppose the model is generating the $(M+1)$-th word on top of $[P_l; \ T_l]$ at the $l$-th inserted layer, we denote the corresponding $(M+1)$-th word token as $t_l \in \mathbb{R}^{1\times C}$. In the attention mechanism, several linear projection layers are first applied to transform the input tokens into queries, keys, and values as

$$Q_l = \text{Linear}_q( \ t_l \ ); \tag{2}$$
$$K_l = \text{Linear}_k( \ [P_l; \ T_l; \ t_l] \ ); \tag{3}$$
$$V_l = \text{Linear}_v( \ [P_l; \ T_l; \ t_l] \ ). \tag{4}$$

Then, the attention scores of $Q_l$ and $K_l$ before the softmax function are calculated as

$$S_l = Q_l K_l^T / \sqrt{C} \ \in \mathbb{R}^{1\times(K+M+1)}, \tag{5}$$

which records the feature similarities between the new word $t_l$ and all $K + M + 1$ tokens. Meanwhile, $S_l$ can be reformulated by two components as

$$S_l = [S_l^K; \ S_l^{M+1}]^T, \tag{6}$$

where $S_l^K \in \mathbb{R}^{K\times 1}$ and $S_l^{M+1} \in \mathbb{R}^{(M+1)\times 1}$ denote the attention scores of $K$ adaption prompts and $M + 1$ word tokens, respectively. The former $S_l^K$ represents how much information the learnable prompt contributes to generating $t_l$, which probably causes disturbance in the early training stage.

Figure 2: **Details of Zero-initialized Attention.** We insert learnable adaption prompts into the last $L$ out of $N$ transformer layers of LLaMA. To progressively learn the instructional knowledge, we adopt a zero gating factor within the attention for stable training in the early training stages.

To this end, we adopt a learnable gating factor, denoted as $g_l$, to adaptively control the importance of $S_l^K$ in the attention. Initialized by zero, $g_l$ can firstly eliminate the influence of under-fitted prompts, and then increase its magnitude for providing more instruction semantics to LLaMA. Therefore, we independently apply the softmax functions to the two components in Equation equation 6, and multiply the first term by $g_l$, formulated as

$$S_l^g = [\text{softmax}(S_l^K) \cdot \tanh(g_l); \ \text{softmax}(S_l^{M+1})]^T, \tag{7}$$

where an activation function $\tanh(\cdot)$ is adopted to regulate the scale of $g_l$ to into -1~1. The separate softmax functions ensure the second term to be irrelevant to the adaption prompts, and we do not multiply any coefficient to $\text{softmax}(S_l^{M+1})$ to prevent the pre-trained knowledge from being disturbed, i.e., preserving its original probability distribution. When $g_l$ is close to zero, it can mostly convey the originally pre-trained knowledge of LLaMA to token $t_l$ for a creditable generation. In practice, we adopt multiple $g_l$ to be independently learned for different heads within the attention, benefiting the learning diversity of multi-head mechanisms.

Finally, we calculate the output of the $l$-th attention layer with a linear projection layer as

$$t_l^o = \text{Linear}_o(S_l^g V_l) \ \in \mathbb{R}^{1\times C}. \tag{8}$$

With our proposed zero-initialized attention, the adaption prompts can progressively inject the newly acquired instructional signals into the transformer, while simultaneously incorporating the pre-trained knowledge of LLaMA to provide high-quality responses.

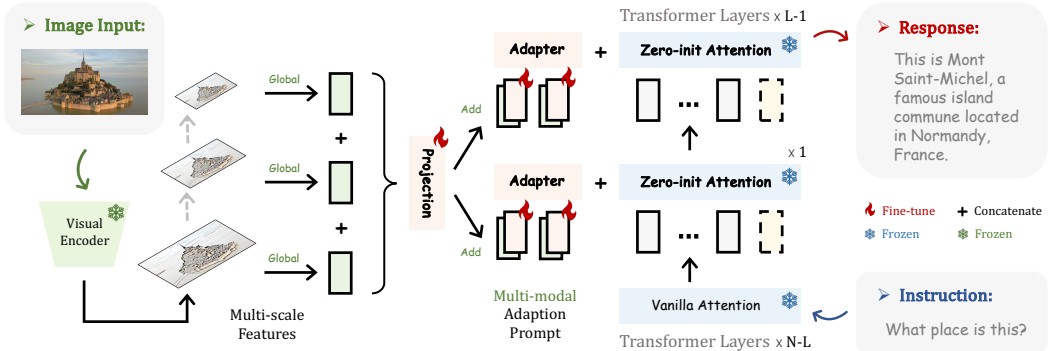

Figure 3: **Multi-modal LLaMA-Adapter.** By connecting a pre-trained image encoder, LLaMA-Adapter can be extended to a multi-modal LLM for image-conditioned instruction following. Given an image input, we element-wisely add the image tokens with adaption prompts, and utilize our zero-initialized attention mechanism to inject visual semantics into LLaMA (Touvron et al., 2023).

## 3.3 Multi-modal Reasoning

Apart from language instructions, LLaMA-Adapter is capable of answering a question based on image input with simple modifications. This fully unleashes the multi-modal reasoning power of LLMs for extensive application scenarios, e.g., image captioning, object counting, and OCR. The overall framework of our multi-modal LLaMA-Adapter is shown in Figure 3.

**Multi-modal Architecture.** For an input image, we first leverage a pre-trained visual encoder, e.g., CLIP (Radford et al., 2021), to extract its multi-scale global features, denoted as $\{I_m\}_{m=1}^M$, where $I_m \in \mathbb{R}^{1 \times C_m}$ and $M$ denotes the scale number. Then, we concatenate the $M$-scale features along the channel dimension, and apply a learnable projection network to transform them into word embedding space, formulated as

$$I_p = \text{Projection}\left(\text{Concat}\left(\{I_m\}_{m=1}^M\right)\right),\tag{9}$$

where $I_p \in \mathbb{R}^{1 \times C}$ and is regarded as the overall image token with the same feature dimension as our adaption prompts. After this, we repeat $I_p$ for $K$ times, and element-wisely add it onto the $K$-length adaption prompts at all $L$ inserted transformer layers. For the $l$-th layer, we denote the acquired multi-modal prompt as

$$P_l^v = P_l + \text{Repeat}(I_p) \ \in \mathbb{R}^{K \times C},\tag{10}$$

where $P_l^v$ denotes the prompt incorporating visual information from the given image. After this, our zero-initialized attention can learn to increasingly infuse the image-conditional semantics into LLaMA by the zero gating factor $g_l$. In this way, an LLM can be efficiently tuned to understand vision-language input, and tackle more challenging generative tasks with multi-modal reasoning.

**Training Strategy.** Instead of using Alpaca's data (Taori et al., 2023) for language-only instruction tuning, we fine-tune LLaMA-Adapter with multi-modal instruction data, and evaluate the performance with two popular scenarios:

- **ScienceQA (Lu et al., 2022) Evaluation.** ScienceQA includes a large-scale science question answering data collected from a wide range of knowledge domains. Each sample contains a visual context, a textual context, a question with multiple options, and an answer. We directly utilize ScienceQA's multi-modal training set to fine-tune LLaMA-Adapter, and conduct in-domain testing. We freeze both the image encoder and LLaMA, and only train the lightweight projection network and zero-initialized attention mechanisms.

- **Zero-shot Multi-modal Evaluation.** To verify the out-of-domain generation ability of our approach, we conduct a two-stage multi-modal training, and then evaluate three benchmarks (MME (Fu et al., 2023), MMBench (Liu et al., 2023c), LVLM-eHub (Xu et al., 2023))

---

**Instruction:** Tell me about alpacas.

**Alpaca:** Alpacas are small, fluffy animals related to camels and llamas. They are native to Peru and Bolivia, and were first domesticated around 5,000 years ago. They are kept mainly for their fine, soft fleece, which is used to make knitwear and other garments. Alpacas are herd animals and live in small family groups. They are highly social animals and can form strong bonds with their owners.

**LLaMA-Adapter:** Alpacas are members of the camelid family and are native to the Andes Mountains of South America. They are typically found in herds of 10-20 animals and are known for their gentle nature and their ability to thrive in harsh climates. Alpacas are typically white or grey in color and have long, silky coats. They are also known for their ability to produce high-quality wool, which is used in the production of clothing and other products.

---

**Instruction:** Translate the sentence 'I have no mouth but I must scream' into Spanish.

**Alpaca:** No tengo boca, pero debo gritar.

**LLaMA-Adapter:** No tengo boca pero tengo que gritar.

---

**Instruction:** Write a Python program that prints the first 10 Fibonacci numbers.

**Alpaca:**

```python
numbers = [0, 1]
for i in range(2, 11):
    numbers.append(numbers[i-2] + numbers[i-1])
print(numbers)
```

**LLaMA-Adapter:**

```python
def print_fibonacci(n):
    a, b = 0, 1
    count = 0
    while count < n:
        print(a, end=' ')
        a, b = b, a + b
        count += 1
print_fibonacci(10)
```

---

Figure 4: **Language Instruction-following Capacity.** Our LLaMA-Adapter performs comparably with Alpaca (Taori et al., 2023) for question answering, language translation, and code generation.

in a zero-shot manner. For the first stage, we utilize the raw image-caption data from LAION-400M (Schuhmann et al., 2021) to tune the projection network and zero-initialized attention modules. This step is mainly for the alignment of embedding space between visual features and word tokens. For the second stage, we freeze the projection network, and only tune the zero-initialized attention within LLaMA by a combination of Alpaca's data and LLaVA-I (Liu et al., 2023b). The second stage aims to endow LLMs with the capability to generate detailed responses based on human instructions.

## 4 EXPERIMENT

In Section 4.1, we first evaluate the language instruction-following capacity of LLaMA-Adapter. Then, we present our multi-modal reasoning performance on several benchmarks in Section 4.2, and conduct ablation studies on ScienceQA's validation set in Section 4.3. Finally, we report the fine-tuning results of our approach on traditional vision and language models in Section 4.4.

### 4.1 INSTRUCTION-FOLLOWING EVALUATION

**Settings.** Following Stanford Alpaca (Taori et al., 2023), we utilize 52K instruction-following data for training. We fine-tune LLaMA-Adapter on 8 A100 GPUs for 5 epochs. The warmup epochs, batch size, learning rate, and weight decay are set to 2, 64, 0.009, and 0.02, respectively. By default, we utilize the pre-trained LLaMA model with 7B parameters and $N = 32$ transformer layers. We

Figure 5: **GPT-4 Evaluating Benchmark** ([Chiang et al., 2023](#)) for LLaMA-Adapter, Alpaca and Alpaca-LoRA.

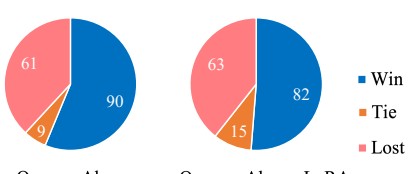

| | Win |
|---|---|
| | Tie |
| | Lost |

Ours vs. Alpaca    Ours vs. Alpaca-LoRA

Table 1: **Efficiency Comparison.** The training time is tested on 8 A100 GPUs.

| Model | Tuned Params | Storage Space | Training Time |
|---|---|---|---|
| Alpaca | 7B | 13G | 3 hours |
| Alpaca-LoRA | 4.2M | 16.8M | 1.5 hours |
| **LLaMA-Adapter** | **1.2M** | **4.7M** | **1 hour** |

Table 2: **Visual Question Answering on ScienceQA** ([Lu et al., 2022](#)) **Benchmark.** $CoT$ denotes using a chain of thought for question answering. $T$ denotes using text-only input.

| Model | Tuned Params | Avg | NAT | SOC | LAN | TXT | IMG | NO | G1-6 | G7-12 |
|---|---|---|---|---|---|---|---|---|---|---|
| Random Choice ([Lu et al., 2022](#)) | - | 39.83 | 40.28 | 46.13 | 29.25 | 47.45 | 40.08 | 33.66 | 39.35 | 40.67 |
| Human ([Lu et al., 2022](#)) | - | 88.40 | 90.23 | 84.97 | 87.48 | 89.60 | 87.50 | 88.10 | 91.59 | 82.42 |
| ChatGPT$_{CoT}$ ([OpenAI, 2023a](#)) | 0M | 78.31 | 78.82 | 70.98 | 83.18 | 77.37 | 67.92 | 86.13 | 80.72 | 74.03 |
| GPT-4$_{CoT}$ ([OpenAI, 2023b](#)) | 0M | 83.99 | 85.48 | 72.44 | 90.27 | 82.65 | 71.49 | 92.89 | 86.66 | 79.04 |
| MCAN ([Yu et al., 2019](#)) | 95M | 54.54 | 56.08 | 46.23 | 58.09 | 59.43 | 51.17 | 55.40 | 51.65 | 59.72 |
| VisualBERT ([Li et al., 2019a](#); [2020](#)) | 111M | 61.87 | 59.33 | 69.18 | 61.18 | 62.71 | 62.17 | 58.54 | 62.96 | 59.92 |
| UnifiedQA ([Khashabi et al., 2020](#)) | 223M | 70.12 | 68.16 | 69.18 | 74.91 | 63.78 | 61.38 | 77.84 | 72.98 | 65.00 |
| UnifiedQA$_{CoT}$ | 223M | 74.11 | 71.00 | 76.04 | 78.91 | 66.42 | 66.53 | 81.81 | 77.06 | 68.82 |
| MM-COT$_T$ ([Zhang et al., 2023e](#)) | 223M | 70.53 | 71.09 | 70.75 | 69.18 | 71.16 | 65.84 | 71.57 | 71.00 | 69.68 |
| MM-COT | 223M | 84.91 | 87.52 | 77.17 | 85.82 | 87.88 | 82.90 | 86.83 | 84.65 | 85.37 |
| LLaMA-Adapter$_T$ | 1.2M | 78.31 | 79.00 | 73.79 | 80.55 | 78.30 | 70.35 | 83.14 | 79.77 | 75.68 |
| LLaMA-Adapter | 1.8M | 85.19 | 84.37 | 88.30 | 84.36 | 83.72 | 80.32 | 86.90 | 85.83 | 84.05 |

adopt a prompt length $K = 10$ and insert the adaption prompts into the last $L = 30$ layers. For quantitative results, we compare with methods both trained by the 52K instruction data, Alpaca ([Taori et al., 2023](#)) and Alpaca-LoRA ([alp, 2023](#)), and evaluate with one widely adopted schemes, GPT-4 evaluating benchmark ([Chiang et al., 2023](#)). It adopts GPT-4 ([OpenAI, 2023b](#)) to assess the quality of two compared responses from different models on 80 questions.

**Performance.** We first show some generated responses of LLaMA-Adapter and Alpaca in Figure 4. For different kinds of instructions, our approach can output reasonable responses comparable to the fully fine-tuned Alpaca, including question answering, language translation, and code generation. Please refer to the Appendix for a full comparison with Alpaca-LoRA, GPT-3 ([Brown et al., 2020](#)), and LLaMA-I ([Touvron et al., 2023](#)). For GPT-4 assessment in Figure 5, LLaMA-Adapter obtains more 'win' compared to Alpaca and Alpaca-LoRA, respectively. This fully demonstrates the effectiveness of our adaption method with zero-initialized attention mechanisms.

**Efficiency.** In Table 1, we compare the learnable parameters, storage space, and training time of different instruction-following methods. As a lightweight plug-and-play module, LLaMA-Adapter enjoys superior training efficiency with only 1.2M parameters, 4.9M storage, and one-hour training. This enables more efficient storage of large-scale language models on mobile devices. LLaMA-Adapter's efficiency advantages can be further revealed by multi-node training, since only the gradients of 1.2M parameters are required to be transferred among nodes, other than Alpaca's 7B.

## 4.2 Multi-modal Evaluation

**Settings.** We adopt CLIP ([Radford et al., 2021](#)) as the image encoder to extract multi-scale visual features, and leverage a simple bottleneck MLP layer as the learnable projection network. We keep other hyperparameters the same as the language instruction-following LLaMA-Adapter. For ScienceQA ([Lu et al., 2022](#)), we concatenate the given question, textual context, and options sequentially in one sentence as LLaMA's input. For zero-shot multi-modal evaluation, we select three benchmarks, MME ([Fu et al., 2023](#)), MMBench ([Liu et al., 2023c](#)), and LVLM-eHub ([Xu et al., 2023](#)), covering a wide range of VQA tasks. We compare with two concurrent multi-modal LLMs: LLaVA ([Liu et al., 2023b](#)) and MiniGPT-4 ([Zhu et al., 2023](#)).

Table 3: **Zero-shot Multi-modal Evaluation** on MME (Fu et al., 2023), MMBench (Liu et al., 2023c) and LVLM-eHub (Xu et al., 2023) benchmarks. P: Perception; C: Cognition. LR: Logical Reasoning; AR: Attribute Reasoning; RR: Relation Reasoning; FP-C/S: Fine-grained Perception (Cross Instance/Single Instance); CP: Coarse Perception. VP: Visual Perception; VKA: Visual Knowledge Acquisition; VR: Visual Reasoning; VC: Visual Commonsense.

| Model | MME | | MMbench | | | | | | | LVLM-eHub | | | |
|---|---|---|---|---|---|---|---|---|---|---|---|---|---|
| | P | C | All | LR | AR | RR | FP-S | FP-C | CP | VP | VKA | VR | VC |
| LLaVA | 503 | 215 | 36.2 | 15.9 | 53.6 | 28.6 | 41.8 | 20.0 | 40.4 | 0.62 | 0.38 | 0.77 | 0.79 |
| Mini-GPT4 | 867 | 292 | 23.0 | 13.6 | 32.9 | 8.9 | 28.7 | 11.2 | 28.3 | 0.73 | 0.35 | 0.53 | 0.57 |
| **LLaMA-Adapter** | 973 | 249 | 39.5 | 13.1 | 47.4 | 23.0 | 45.0 | 33.2 | 50.6 | 0.81 | 0.44 | 0.83 | 0.59 |

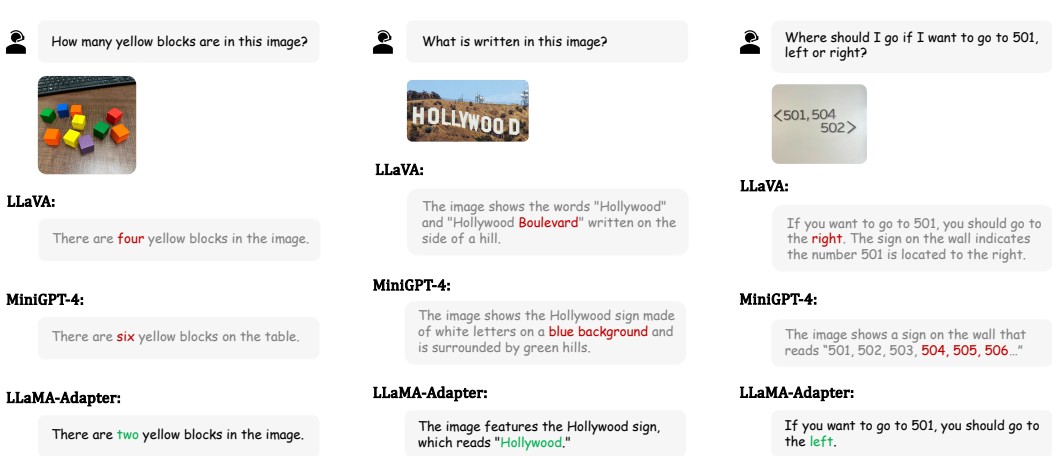

Figure 6: **Multi-modal Reasoning Examples** on zero-shot open-domain questions. We compare our approach with LLaVA (Liu et al., 2023b) and MiniGPT-4 (Zhu et al., 2023) for object counting, OCR, and common sense reasoning.

**Performance.**    In Table 2 for the ScienceQA performance, our single-modal 'LLaMA-Adapter$_T$' attains 78.31% accuracy, surpassing several traditional VQA methods with large parameters. By further injecting visual conditions with a 0.6M projection network, our multi-modal 'LLaMA-Adapter' improves +6.88% accuracy, attaining leading results superior to the GPT series. In Table 3 for the three multi-modal benchmarks, compared to the concurrent works, our approach achieves competitive scores with a much more efficient tuning strategy. This is because, LLaVA requires fine-tuning the entire 7B LLM, and Mini-GPT4 adopts Vicuna (Chiang et al., 2023) that also fully fine-tunes LLaMA with 13B parameters. We also show some multi-modal reasoning examples in Figure 6. Our approach exhibits better object counting, OCR, and commence reasoning performance.

### 4.3    ABLATION STUDY

**Insertion Layers.**    We first investigate the number of transformer layers to be inserted by zero-initialized attention in LLaMA-Adapter. As shown in Table 4, increasing the layer numbers introduces more parameters, but leads to a large improvement in the answering accuracy of ScienceQA's validation set. There also exists an optimal insertion number from the higher layers, since too many layers would adversely disturb the early encoding of input words. If one has limited resources to identify the best number, simply inserting into all transformer layers is generally a good solution.

**Zero-initialized Attention.**    Our proposed zero-initialized attention is essential for the early-stage training stability and final generation capacity. As shown in Table 5, it contributes to a significant +43.08% gain on ScienceQA's validation set. In contrast, the randomly initialized baseline only achieves 40.77% accuracy, nearly the same as 'Random Choice' (Table 2's first row). In Figure 7, we plot the loss curves with and without the zero initialization. The loss of 'Zero-initialized' declines much faster at the beginning, and finally converges to zero. In contrast, the 'Random-initialized' slowly approaches 0.15, which is not fully converged and causes a large performance drop.

Table 4: **Number of Insertion Layers** to the pre-trained transformer of LLaMA.

| Layers | Params | Val Acc. |
|--------|--------|----------|
| 10 | 0.97 | 55.95 |
| 20 | 1.37 | 73.36 |
| 30 | 1.79 | **83.85** |
| 32 | 1.83 | 81.03 |

Table 5: Effectiveness of **Zero-initialized Attention** in our method.

| Setting | Val Acc. |
|---------|----------|
| Rand-Init. | 40.77 |
| Zero-Init. | **83.85** |
| *Gain* | +43.08 |

Figure 7: **Loss Curves** of LLaMA-Adapter with (blue) and without (orange) zero-initialized attention.

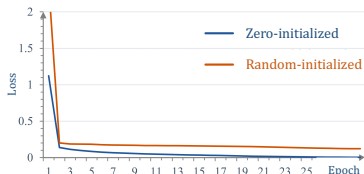

Table 6: **Vision Model Fine-tuning** with ViT on VTAB-1k benchmark.

| Method | Natural | Special. | Struct. |
|--------|---------|----------|---------|
| Full | 75.88 | 83.36 | 47.64 |
| Adapter | 70.39 | 77.11 | 33.43 |
| Sidetune | 58.21 | 68.12 | 23.41 |
| VPT | 78.48 | 82.43 | 54.98 |
| **Zero-init.** | **81.74** | **84.43** | **56.75** |

Table 7: **Language Model Fine-tuning** with RoBERTa on SQuAD benchmark.

| Method | SQuAD 1.1 | | SQuAD 2.0 | |
|--------|-----|-----|-----|-----|
| | EM | F1 | EM | F1 |
| Full | 88.9 | 94.6 | 86.5 | 89.4 |
| PT | 1.2 | 12.0 | 50.2 | 50.2 |
| PT2 | 88.5 | 94.4 | 82.1 | 85.5 |
| **Zero-init.** | **88.8** | **94.6** | **83.9** | **87.2** |

Table 8: **Vision-language Fine-tuning** with CLIP on base-to-novel benchmark.

| Method | Base | Novel | HM |
|--------|------|-------|-----|
| CLIP | 75.88 | 83.36 | 47.64 |
| CoOp | 70.39 | 77.11 | 33.43 |
| CoCop | 58.21 | 68.12 | 23.41 |
| MaPLe | 78.48 | 82.43 | 54.98 |
| **Zero-init.** | **81.74** | **84.43** | **56.75** |

## 4.4 Zero-initialized Attention for other Large Models

Our approach, i.e., zero-initialized attention, is not limited to the domain of tuning instruction models, and can be further utilized to fine-tune large models in traditional vision and language tasks.

**Vision Models.** We select a pre-trained ViT/16 (Dosovitskiy et al., 2020) as the vision model and evaluate on VTAB-1k (Zhai et al., 2019) benchmark, which contains 19 visual tasks with three domains: Natural, Specialized, and Structured. As shown in Table 6, for various image distributions, e.g., natural images, medical and satellite imagery, our approach performs much better than the full fine-tuning, and also surpasses existing parameter-efficient methods (Jia et al., 2022; Houlsby et al., 2019; Zhang et al., 2020), indicating our generalization ability for vision tasks.

**Language Models.** We utilize a pre-trained RoBERTa_large (Liu et al., 2019) and adopt SQuAD (Rajpurkar et al., 2016) v1.1 and v2.0 benchmarks for extractive question answering evaluation. Exact Match (EM) and F1 scores on the dev set are reported. We refer to the Appendix for other language tasks. As shown in Table 7, the leading results among previous methods (Lester et al., 2021; Liu et al., 2021a) demonstrate our superiority over traditional language tasks.

**Vision-language Models.** We adopt CLIP (Radford et al., 2021) as the pre-trained vision-language model, and test on base-to-novel generalization (Zhou et al., 2022b) benchmark, where 'HM' denotes harmonic mean. As shown in Table 8, compared to previous works (Zhou et al., 2022a;c; Khattak et al., 2022), our approach achieves the best average classification accuracy on both base and novel categories, demonstrating our fine-tuning capability for large vision-language models.

## 5 Conclusion

In this paper, we propose LLaMA-Adapter, an efficient adaption method for tuning instruction-following models. For better training stability and final performance, we introduce the zero-initialized attention mechanism with a learnable gating factor, which increasingly incorporates instructional signals, while preserving the pre-trained knowledge in LLaMA. With only 1.2M parameters and one-hour training, our approach effectively fine-tunes LLaMA with superior efficiency compared to the 7B-parameter Alpaca. LLaMA-Adapter can be generalized to image-conditioned generation as a multi-modal LLM, achieving competitive results on various visual question answering benchmarks. On traditional vision and language tasks, our zero-initialized attention also attains favorable fine-tuning performance, which indicates strong generalization capacity.

## 6 ACKNOWLEDGEMENT

This work is partially supported by the National Key R&D Program of China (NO.2022ZD0161100), the National Natural Science Foundation of China (No.62206272), the Centre for Perceptual and Interactive Intelligence (CPII) Ltd under the Innovation and Technology Commission (ITC)'s InnoHK, and General Research Fund of Hong Kong RGC Project 14204021. Hongsheng Li is a PI of CPII under the InnoHK.

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

# A  OVERVIEW

# B  MORE DETAILS OF MULTI-MODAL EVALUATION

**ScienceQA (Lu et al., 2022) Evaluation.**  The data sample in ScienceQA contains a visual context, a textual context, a question, multiple options, and a correct answer, as shown in Figure 8. We omit the lecture and explanation in some data samples for simplicity.

**Zero-shot Multi-modal Evaluation.**  We test our approach on the three benchmarks (Fu et al., 2023; Liu et al., 2023c; Xu et al., 2023) following their official procedures. In Tables 9, 10 and 11, we respectively report the detailed results of MME and LVLM-eHub benchmarks. As shown, across a wide range of visual question-answering problems, our approach can consistently achieve competitive results. We also show more examples of the multi-modal LLaMA-Adapter for open-domain zero-shot visual questions in Figure 9, where our approach can generate detailed and high-quality responses in natural language. The experiments fully demonstrate the generalization capacity of our proposed multi-modal LLM. We also give some qualitative examples in Figures 9 and 10, where our LLaMA-Adapter can answer open-ended questions for web images.

Table 9: **Perception Results on MME Benchmark (Fu et al., 2023).**

| Model | ALL | Existence | Count | Position | Color | Poster | Celebrity | Scene | Landmark | Artwork | OCR |
|---|---|---|---|---|---|---|---|---|---|---|---|
| LLaVA | 503 | 50 | 50 | 50 | 55 | 50 | 49 | 50 | 50 | 49 | 50 |
| MiniGPT-4 | 867 | 115 | 123 | 82 | 110 | 56 | 65 | 96 | 69 | 56 | 83 |
| LLaMA-Adapter | 973 | 120 | 50 | 48 | 75 | 100 | 86 | 149 | 150 | 70 | 125 |

Table 10: **Cognition Results on MME Benchmark (Fu et al., 2023).**

| Model | ALL | Commonsense Reasoning | Numerical Calculation | Text Translation | Code Reasoning |
|---|---|---|---|---|---|
| LLaVA | 215 | 57 | 50 | 58 | 50 |
| MiniGPT-4 | 292 | 72 | 55 | 55 | 110 |
| LLaMA-Adapter | 249 | 81 | 63 | 50 | 55 |

Table 11: **Zero-shot Multi-modal Results on LVLM-eHub Benchmark (Xu et al., 2023).** OC: Object Counting; MCI: Multi-Class Identification; KIE: Key Information Extraction; VE: Visual Entailment; KGID: Knowledge-grounded Image Description; VCR: Visual Commonsense Reasoning.

| LVLM-eHub | Tasks | #Datasets | LLaVA | MiniGPT-4 | LLaMA-Adapter |
|---|---|---|---|---|---|
| Visual Perception | ImgCls, OC, MCI | 8 | 0.62 | 0.73 | 0.81 |
| Visual Knowledge Acquisition | OCR, KIE, Caption | 17 | 0.38 | 0.35 | 0.44 |
| Visual Reasoning | VQA, KGID, VE | 13 | 0.77 | 0.53 | 0.83 |
| Visual Commonsense | ImageNetVC, VCR | 6 | 0.79 | 0.57 | 0.59 |
| Average | - | 44 | 0.64 | 0.55 | 0.67 |

# C  ADDITIONAL RELATED WORK

**Multi-modal Language Models.**  With the continuous improvement of data scale and computing power, the advancement of Multi-Modal Language Models (MMLMs) has gained momentum.

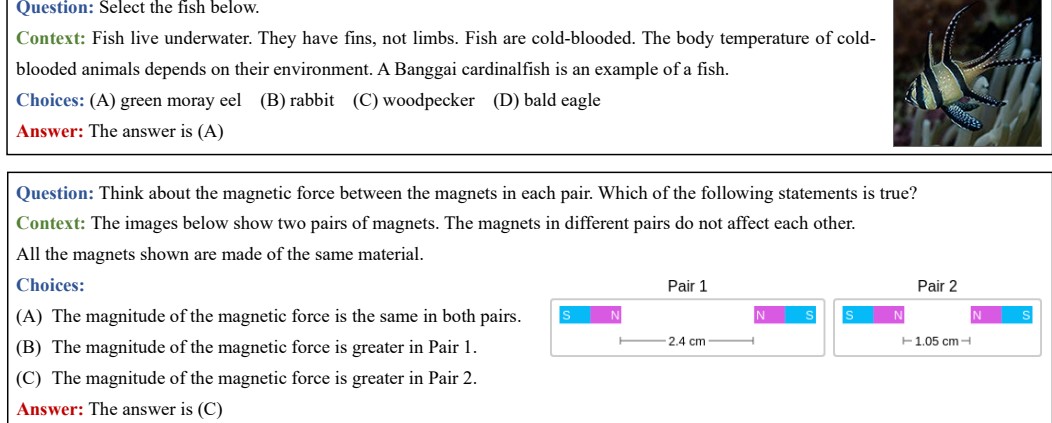

Figure 8: **Multi-modal Reasoning Examples in ScienceQA (Lu et al., 2022).**

Initiatives like CLIP (Radford et al., 2021), ALIGN (Jia et al., 2021), and their derivatives (Li et al., 2022; Gao et al., 2021; Zhai et al., 2022) employ vision-language contrastive pre-training on vast datasets, showcasing robust generalization in zero-shot evaluation. With the rise of LLMs (OpenAI, 2023a;b), modern MMLMs merge these LLM architectures with visual comprehension capacities. BLIP-2 (Li et al., 2023b), for instance, introduces a Q-Former network, bridging frozen image encoders with LLMs. Flamingo (Alayrac et al., 2022) uses interleaved image-text data for few-shot learning, enhancing vision-language inferences. Kosmos (Huang et al., 2023) trains an MMLM on web-scale multi-modal data from scratch, enabling powerful visual perception capacities. While models like Bard (Google, 2023) and GPT-4 (OpenAI, 2023b) remain influential, their closed-source nature has led to the development of MMLMs such as those based on open-source LLaMA (Liu et al., 2023b; Zhu et al., 2023; Ye et al., 2023; Li et al., 2023a; Zhang et al., 2023a). Typically, these MMLMs utilize a two-stage training process. In the initial phase, a substantial quantity of image-text pairs are leveraged to align vision models with LLMs. The subsequent phase involves fine-tuning on a limited set of high-quality datasets to follow human instructions. However, these models are either highly dependent on a fine-tuned instruction model (Vicuna (Chiang et al., 2023) in Mini-GPT4 (Zhu et al., 2023)), or require updating the entire parameters of LLMs (LLaVA (Liu et al., 2023b)). Follow-up works like SPHINX series (Lin et al., 2023; Gao et al., 2024), ImageBind-LLM (Han et al., 2023), Point-LLM (Guo et al., 2023), and MathVerse (Zhang et al., 2024) further explore more diverse potentials of multi-modal capabilities within LLMs. As a concurrent work to LLaVA and Mini-GPT4, our LLaMA-Adapter utilizes zero-initialized attention mechanisms for parameter-efficiently fine-tuning the original LLaMA model, largely saving the expensive full-parameter fine-tuning.

**Comparison to Flamingo (Alayrac et al., 2022).** As a strong in-context MLLMs, Flamingo adopts a gating strategy for injecting external knowledge into LLMs. Compared to our zero-initialized attention, there are three main differences as follows.

- **Inserted Position.** Our gating works delicately within the self-attention layer of an LLM, more specifically, after the query-key attention scores and before multiplying with value. In contrast, Flamingo's gating is outside and before feeding into LLM's layers, which works right after the newly added cross-attention layer and feed-forward networks.

- **Detailed Mechanism.** Our gating directly reweighs the attention scores of adaption prompts, controlling how much information of prompts is aggregated by the generating word token. Flamingo's gating naively reweighs the residual connection, which controls how much information of visual features is added to all language features.

- **Parameter Efficiency.** Our gating mechanism only introduces 1.2M parameters of efficient learnable prompts. Flamingo's gating is based on newly added large-scale cross-attention layers and FFNs, having over 3B parameters.

- **Application Scenarios.** Due to our lightweight designs, the zero-initialized attention can be adopted either for incorporating language instruction knowledge, or multi-modal image conditions, while Flamingo is specially designed for vision-language tasks by newly adding heavyweight cross-attention modules.

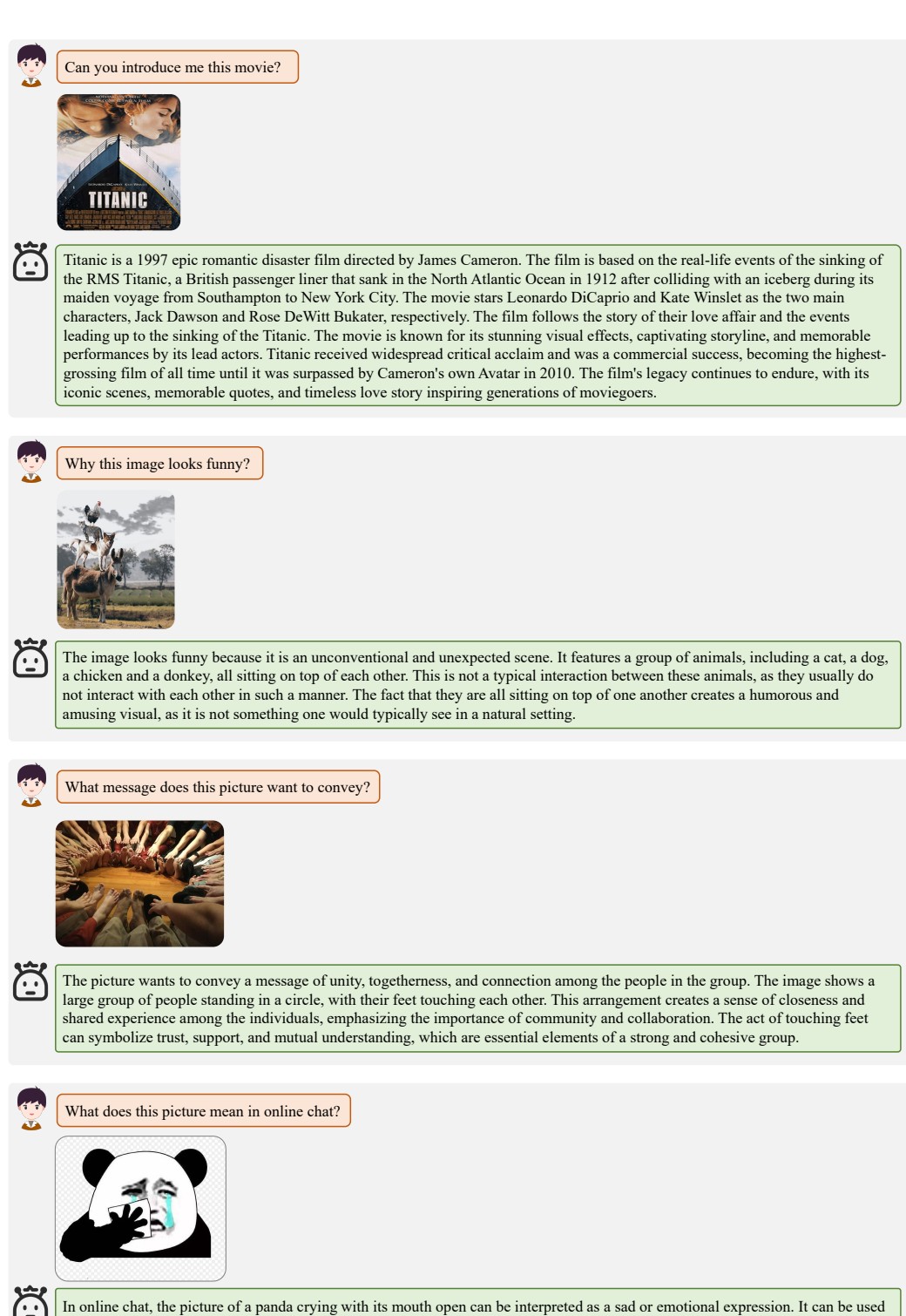

Figure 9: **Zero-shot Multi-modal Understanding Examples of LLaMA-Adapter: Part 1.**

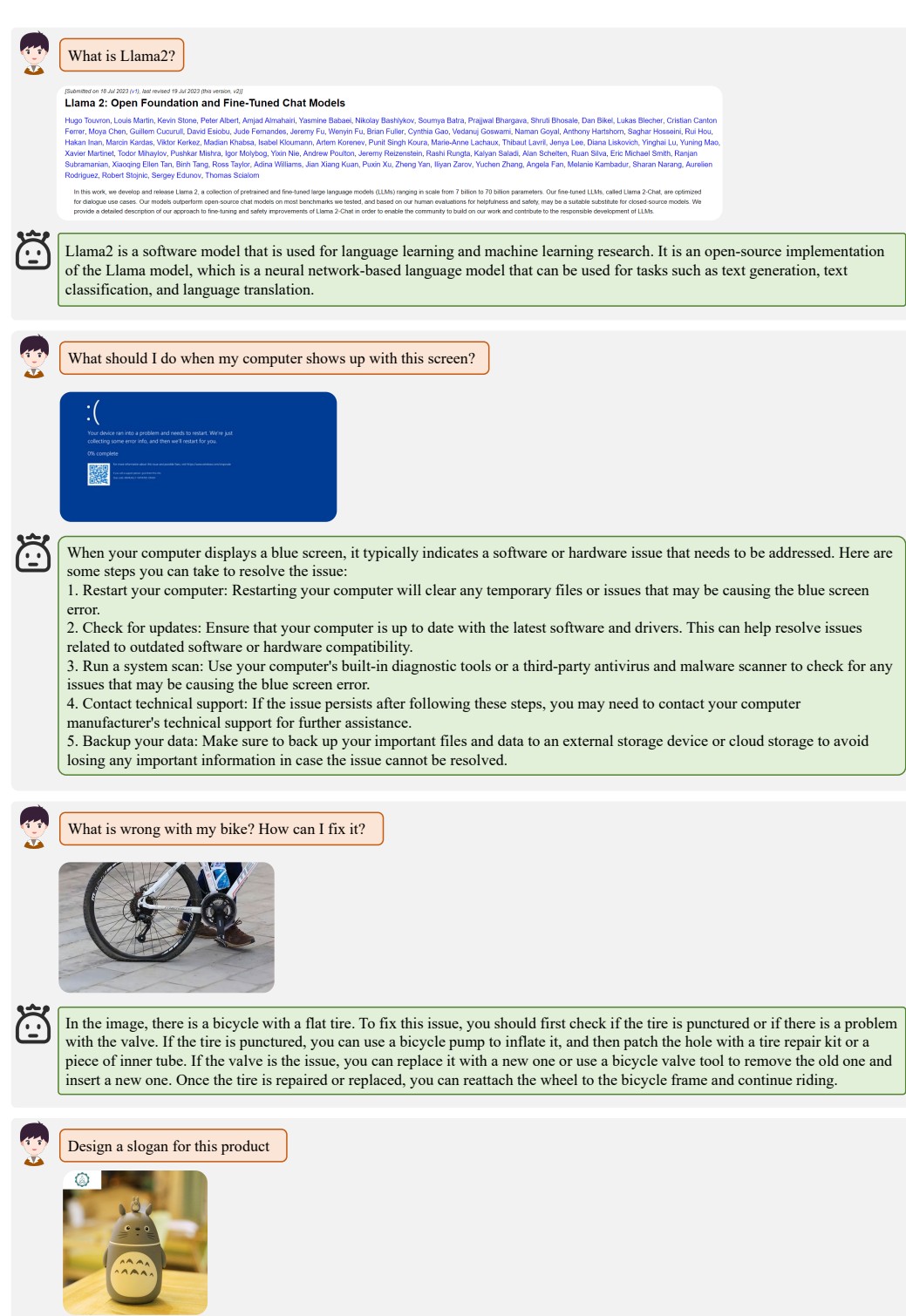

Figure 10: **Zero-shot Multi-modal Understanding Examples of LLaMA-Adapter: Part 2.**

# D  MORE DETAILED RESULTS OF MODEL FINE-TUNING

In this section, we provide more detailed experiments and analysis of applying our zero-initialized attention to fine-tune vision models, language models, and vision-language models, respectively.

## D.1  DETAILED RESULTS ON VISION MODELS

In Table 12, we compare the detailed fine-tuning results on VTAB-1k (Zhai et al., 2019) benchmark with 19 downstream visual tasks, which can be categorized into Natural (7 tasks), Specialized (4 tasks), and Structured (8 tasks), according to image domains. As shown, our zero-initialized attention outperforms VPT (Jia et al., 2022) on most datasets (16 out of 19), and surpasses full fine-tuning along with other fine-tuning methods by large margins. This demonstrates the general efficacy of the proposed mechanism on a variety of image distributions.

Table 12: **Detailed Fine-tuning Results on VTAB-1k Benchmark.** We report the top-1 accuracy and adopt ViT-B/16 (Dosovitskiy et al., 2020) pre-trained on supervised ImageNet-21k (Deng et al., 2009) as the base model. We compare our zero-initialized attention with Bias (Zaken et al., 2022), Adapter (Houlsby et al., 2019), Sidetune (Zhang et al., 2020) and VPT (Jia et al., 2022).

| | CIFAR100 | Caltech101 | DTD | Flowers102 | OxfordPets | SVHN | SUN397 | Mean | Patch Camelyon | EuroSAT | Resisc-45 | Retinopathy | Mean | Clevr/count | Clevr/distance | DMLab | KITTI/distance | dSprites/location | dSprites/orientation | SmallNORB/azimuth | SmallNORB/elevation | Mean |
|---|---|---|---|---|---|---|---|---|---|---|---|---|---|---|---|---|---|---|---|---|---|---|
| Full | 68.9 | 87.7 | 64.3 | 97.2 | 86.9 | 87.4 | 38.8 | 75.9 | 79.7 | 95.7 | 84.2 | 73.9 | 83.4 | 56.3 | 58.6 | 41.7 | 65.5 | 57.5 | 46.7 | 25.7 | 29.1 | 47.6 |
| Bias | 72.8 | 87.0 | 59.2 | 97.5 | 85.3 | 59.9 | 51.4 | 73.3 | 78.7 | 91.6 | 72.9 | 69.8 | 78.3 | 61.5 | 55.6 | 32.4 | 55.9 | 66.6 | 40.0 | 15.7 | 25.1 | 44.1 |
| Adapter | 74.1 | 85.7 | 62.7 | 97.8 | 87.2 | 34.6 | 50.7 | 70.4 | 76.3 | 87.5 | 73.7 | 70.9 | 77.1 | 45.2 | 41.8 | 31.2 | 56.4 | 31.9 | 25.4 | 13.5 | 22.0 | 33.4 |
| Sidetune | 60.7 | 60.8 | 53.6 | 95.5 | 66.7 | 34.9 | 35.3 | 58.2 | 58.5 | 87.7 | 65.2 | 61.0 | 68.1 | 27.6 | 22.6 | 31.3 | 51.7 | 8.2 | 14.4 | 9.8 | 21.8 | 23.4 |
| VPT | 78.8 | 90.8 | 65.8 | 98.0 | 88.3 | 78.1 | 49.6 | 78.5 | 81.8 | 96.1 | 83.4 | 68.4 | 82.4 | 68.5 | 60.0 | 46.5 | 72.8 | 73.6 | 47.9 | 32.9 | 37.7 | 55.0 |
| **Zero-init.** | 82.2 | 92.4 | 70.3 | 98.4 | 89.8 | 84.9 | 54.3 | **81.7** | 83.6 | 95.3 | 85.0 | 73.8 | **84.4** | 69.3 | 60.2 | 51.1 | 79.7 | 80.7 | 49.0 | 30.6 | 33.6 | **56.8** |

## D.2  MORE EXPERIMENTS ON LANGUAGE TASKS

For a more comprehensive evaluation of zero-initialized attention, we fine-tune RoBERTa$_{large}$ (Liu et al., 2019) on other two natural language processing tasks in addition to extractive question answering of the main paper, which are named entity recognition (NER) and semantic role labeling (SRL). We adopt CoNLL03 (Sang & De Meulder, 2003), CoNLL04 (Carreras & Màrquez, 2004), CoNLL05 (Carreras & Màrquez, 2005), and CoNLL12 (Pradhan et al., 2012) as the evaluation datasets. As shown in Table 13, compared to P-tuning V2 (PT2) (Liu et al., 2021a), our zero-initialized attention can steadily perform better on all datasets with varying magnitudes, which indicates our effectiveness for different language tasks and applications.

Table 13: **Language Model Fine-tuning** with RoBERTa$_{large}$ (Liu et al., 2019) on named entity recognition (NER) and semantic role labeling (SRL) tasks. We report the micro-f1 score. * denotes our reproduced results.

| Method | CoNLL03 | CoNLL04 | CoNLL12 | CoNLL05$_{Brown}$ | CoNLL05$_{WSJ}$ |
|---|---|---|---|---|---|
| Full | 92.6 | 88.8 | 86.5 | 85.6 | 90.2 |
| PT (Lester et al., 2021) | 86.1 | 76.2 | 67.2 | 70.7 | 76.8 |
| PT2 (Liu et al., 2021a) | 92.8 | 88.4 | 84.6 | 84.3 | 89.2 |
| PT2* | 91.8 | 88.4 | 84.7 | 83.9 | 89.4 |
| **Zero-init.** | **92.4** | **88.8** | **85.2** | **84.7** | **89.6** |

## D.3  DETAILED RESULTS ON VISION-LANGUAGE MODELS

Besides ViT and RoBERTa, we also evaluate our approach on CLIP (Radford et al., 2021), a vision-language model pre-trained by 400 million text-image pairs. In detail, we adopt CLIP with a ViT-B/16 as the visual encoder and a 12-layer transformer (Li et al., 2019b) as the textual encoder. We test our

Table 14: **Vision-Language Model Fine-tuning** with ViT-B/16 CLIP (Radford et al., 2021) on base-to-novel generalization (Zhou et al., 2022b) benchmark. We report the classification accuracy (%) and harmonic mean (HM).

| Method | ImageNet | | | Caltech101 | | | Flowers102 | | | Average | | |
|---|---|---|---|---|---|---|---|---|---|---|---|---|
| | Base | Novel | HM | Base | Novel | HM | Base | Novel | HM | Base | Novel | HM |
| CLIP (Radford et al., 2021) | 72.43 | 68.14 | 70.22 | 96.84 | 94.00 | 95.40 | 72.08 | 77.80 | 74.83 | 80.45 | 79.98 | 80.15 |
| CoOp (Zhou et al., 2022c) | 76.47 | 67.88 | 71.92 | 98.00 | 89.81 | 93.73 | 97.60 | 59.67 | 74.06 | 90.69 | 72.45 | 79.90 |
| CoCoOp (Zhou et al., 2022b) | 75.98 | 70.43 | 73.10 | 97.96 | 93.81 | 95.84 | 94.87 | 71.75 | 81.71 | 89.60 | 78.66 | 83.55 |
| MaPLe (Khattak et al., 2022) | 76.66 | 70.54 | 73.47 | 97.74 | 94.36 | 96.02 | 95.92 | 72.46 | 82.56 | 90.11 | 79.12 | 84.02 |
| **Zero-init.** | **76.70** | **71.00** | **73.74** | **98.10** | **94.53** | **96.28** | **96.00** | **74.67** | **84.00** | **90.27** | **80.07** | **84.67** |

fine-tuning results on base-to-novel generalization (Zhou et al., 2022b) benchmark with three datasets, i.e., ImageNet (Deng et al., 2009), Caltech101 (Fei-Fei et al., 2004), and Flowers102 (Nilsback & Zisserman, 2008), where the model is trained only on the base classes in a few-shot setting and evaluated on both base and novel categories. We freeze the entire CLIP and insert the adaption prompts with zero-initialized attention into CLIP's encoders. As shown in Table 14, our approach achieves the best average classification accuracy on both base and novel categories, demonstrating our fine-tuning capability for large vision-language models.

# E    ADDITIONAL EXPERIMENT AND DISCUSSION

## E.1    EVALUATION ON COUNTERFACTUAL REASONING

As a core ability of human intelligence, counterfactual reasoning is a challenging assessment for multi-modal LLMs, which involves the processing of alternatives to observed states or past events. Here, we adopt the very recent C-VQA (Zhang et al., 2023b) benchmark for evaluating our counterfactual reasoning capability. C-VQA contains 2K counterfactual question and answer pairs, which are collected from VQAv2 (Goyal et al., 2017) and supplemented by ChatGPT (OpenAI, 2023a). As shown in Table 15, for three groups of questions, LLaMA-Adapter performs comparably to the concurrent LLaVA. Especially for the 'Numerical indirect' questions, our approach achieves the best counterfactual reasoning results (34.3) and the least performance loss (5.6↓) than all other models.

Table 15: **Counterfactual Reasoning Evaluation on C-VQA (Zhang et al., 2023b) Benchmark.**

| Method | Numerical direct↑ (Loss↓) | Numerical indirect↑ (Loss↓) | Boolean↑ (Loss↓) |
|---|---|---|---|
| ViperGPT (Surís et al., 2023) | 80.6 (19.4↓) | 31.6 (68.4↓) | 21.6 (72.4↓) |
| LLaVA-7B (Liu et al., 2023b) | 27.0 (9.9↓) | 25.0 (15.2↓) | 58.5 (4.8↓) |
| LLaVA-13B (Liu et al., 2023b) | 24.8 (11.9↓) | 20.8 (21.2↓) | 56.3 (4.7↓) |
| LLaMA-Adapter-7B | 30.1 (5.8↓) | 34.3 (5.6↓) | 45.8 (14.5↓) |

## E.2    EVALUATION ON OBJECT HALLUCINATION

Similar to language generation, multi-modal LLMs also suffer from the hallucination issue, i.e., they might generate descriptions containing objects inconsistent with the target images. To validate our approach's performance, we adopt POPE (Li et al., 2023c) for object hallucination evaluation, which converts the object hallucination problem as a binary classification task and includes 500 images from MSCOCO (Lin et al., 2014) with 6 questions per sample. As shown in Table 16, for different evaluation settings, LLaMA-Adapter with LLaMA-7B attains competitive accuracy compared to other multi-modal LLMs with LLaMA-13B, which indicates our relatively stronger robustness to object hallucination problems.

## E.3    TUNING BY MORE INSTRUCTION DATA

By default, we utilize a combination of Alpaca's data (52K) (Taori et al., 2023) and LLaVA-I (158K) Liu et al. (2023b) for visual instruction tuning. Here, we progressively add more

Table 16: **Object Hallucination Evaluation on POPE (Li et al., 2023d) Benchmark.**

| Method | Random | Popular | Adversarial |
|---|---|---|---|
| InstructBLIP-13B (Dai et al., 2023b) | 88.73 | 81.37 | 74.37 |
| mPLUG-Owl-7B (Ye et al., 2023) | 53.30 | 50.63 | 50.67 |
| LLaVA-13B (Liu et al., 2023b) | 54.43 | 52.43 | 50.77 |
| MM-GPT-7B (Gong et al., 2023) | 50.03 | 50.00 | 50.00 |
| LLaMA-Adapter-7B | 75.47 | 60.43 | 60.66 |

question-answering data to enlarge the instruction datasets of LLaMA-Adapter: the sampled 83K VQAv2 (Goyal et al., 2017) by LLaVA-1.5 (Liu et al., 2023a) and the entire 204K VQAv2. We also compare our performance with very recent multi-modal LLMs with advanced visual reasoning capabilities: InstructBLIP (Dai et al., 2023a) and LLaVA-1.5. InstructBLIP collects extensive visual question-answering datasets (16M) to fine-tune BLIP-2 (Li et al., 2023b), which endows robust visual instruction-following capabilities. LLaVA-1.5 is an upgraded variant of LLaVA with a more powerful LLM, i.e., LLaMA-2 (Touvron et al., 2023), and is also fine-tuned by a collection of 665K instruction-tuning datasets. As shown in Table 17, the increasing instruction tuning data leads to better multi-modal reasoning results on three benchmarks, demonstrating our method's favorable scalability to data size. Our LLaMA-Adapter also achieves comparable performance to the latest InstructBLIP and LLaVA-1.5, further indicating our effectiveness for multi-modal reasoning.

Table 17: **Instruction-tuning with More Datasets** on three zero-shot multi-modal Benchmarks: MME (Fu et al., 2023), MMBench (Liu et al., 2023c), and LVLM-eHub (Xu et al., 2023).

| Model | MME | | | MMBench | | | | | | | LVLM-eHub | | | | |
|---|---|---|---|---|---|---|---|---|---|---|---|---|---|---|---|
| | All | P | C | All | LR | AR | RR | FP-S | FP-C | CP | All | VP | VKA | VR | VC |
| BLIP-2 | 1584 | 1294 | 290 | - | - | - | - | - | - | - | 0.77 | 0.86 | 0.93 | 0.76 | 0.54 |
| InstructBLIP | 1505 | 1213 | 292 | 33.9 | 21.6 | 47.4 | 22.5 | 33.0 | 24.4 | 41.1 | 0.95 | 0.93 | 0.97 | 0.91 | 0.99 |
| MiniGPT-4 | 1159 | 867 | 292 | 23.0 | 13.6 | 32.9 | 8.9 | 28.7 | 11.2 | 28.3 | 0.55 | 0.73 | 0.35 | 0.53 | 0.57 |
| LLaVA | 718 | 503 | 215 | 36.2 | 15.9 | 53.6 | 28.6 | 41.8 | 20.0 | 40.4 | 0.64 | 0.62 | 0.38 | 0.77 | 0.79 |
| LLaVA-1.5 | 1826 | 1531 | 295 | 59.5 | 32.4 | 72.6 | 49.3 | 62.3 | 52.2 | 67.7 | - | - | - | - | - |
| LLaMA-Adapter | 1222 | 973 | 249 | 39.5 | 13.1 | 47.4 | 23.0 | 45.0 | 33.2 | 50.6 | 0.6675 | 0.81 | 0.44 | 0.83 | 0.59 |
| +VQAv2 (83K) | 1256 | 1007 | 249 | 43.4 | 22.9 | 44.7 | 31.3 | 46.7 | 46.9 | 50.3 | 0.6925 | 0.84 | 0.42 | 0.88 | 0.63 |
| +VQAv2 (204K) | 1618 | 1272 | 346 | 60.1 | 34.7 | 65.3 | 48.7 | 63.1 | 57.3 | 69.3 | 0.7175 | 0.86 | 0.44 | 0.92 | 0.65 |

## E.4 More Quantitative Comparison with Alpaca-LoRA

Besides qualitative results, We have compared the language generative capabilities of our LLaMA-Adapter, Alpaca (Taori et al., 2023), and Alpaca-LoRA (alp, 2023) on the GPT-4 evaluation benchmark (Chiang et al., 2023) in Figure 5, which utilizes GPT-4 to assess the response quality on 80 questions. Here, we further evaluate the language processing capacity of the three methods on Open LLM benchmark (Edward Beeching, 2023). It evaluates LLMs' generative abilities in four different tasks: AI2 Reasoning Challenge (Clark et al., 2018), HellaSwag (Zellers et al., 2019), MMLU (Hendrycks et al., 2021), and TruthfulQA (Lin et al., 2022). Each task contains challenging data samples over a wide range of knowledge domains. As shown in Table 18, LLaMA-Adapter still achieves the best average performance than Alpaca's full fine-tuning and Alpaca-LoRA. This demonstrates the strong language instruction-following ability of our approach.

Table 18: **Quantitative Evaluation on Open LLM Benchmark (Edward Beeching, 2023).**

| Method | Avg | ARC | HellaSwag | MMLU | TruthfulQA |
|---|---|---|---|---|---|
| Alpaca (Taori et al., 2023) | 49.23 | 49.1 | 77.7 | 33.8 | 36.3 |
| Alpaca-LoRA (alp, 2023) | 50.73 | 53 | 77.9 | 37.1 | 34.9 |
| LLaMA-Adapter | 52.2 | 54.7 | 78.8 | 34.9 | 40.4 |

### E.5 Comparison to Different LoRA Variants

The default rank of Alpaca-LoRA (alp, 2023) is 8, which contains 4.2M trainable parameters. In Table 19, we respectively show the results of Alpaca-LoRA with the ranks of 2, 4, and 16. We also evaluate their language processing capabilities on Open LLM benchmark (Edward Beeching, 2023). As shown, lower ranks of LoRA can effectively reduce the learnable parameters from 8.4M to 1.0M, and slightly lower the training time from 1.5h to 1.48h. However, our LLaMA-Adapter with 1.2M parameters and 1h still attains the best average result, demonstrating a good trade-off between performance and training efficiency.

Table 19: **Comparison to Alpaca-LoRA (alp, 2023) with Different Ranks** on Open LLM benchmark (Edward Beeching, 2023).

| Model | Rank | Param | Time | AVG | ARC | HellaSwag | MMLU | TruthfulQA |
|---|---|---|---|---|---|---|---|---|
| Alpaca-LoRA | 2 | 1.0 | 1.48 | 50.9 | 53.6 | 77.9 | 37.9 | 34.0 |
| | 4 | 2.1 | 1.49 | 50.8 | 53.5 | 77.8 | 37.5 | 34.4 |
| | 8 | 4.2 | 1.49 | 50.7 | 53.2 | 78.1 | 37.1 | 34.5 |
| | 16 | 8.4 | 1.5 | 50.8 | 53.0 | 78.0 | 37.1 | 34.9 |
| LLaMA-Adapter | - | 1.2 | 1.0 | 52.2 | 54.7 | 78.8 | 34.9 | 40.4 |

## F  Full Comparison of Instruction-following Models

In this section, we provide the full comparison of existing instruction-following models: Alpaca (Taori et al., 2023), Alpaca-LoRA (alp, 2023), GPT-3 (Brown et al., 2020), and our LLaMA-Adapter. Our approach only fine-tunes 1.2M parameters within one hour, but generates responses comparable to the fully fine-tuned Alpaca and large-scale GPT-3, exhibiting a superior performance-efficiency trade-off.

---

**Instruction:** Tell me about alpacas.

**LLaMA-Adapter:** Alpacas are members of the camelid family and are native to the Andes Mountains of South America. They are typically found in herds of 10-20 animals and are known for their gentle nature and their ability to thrive in harsh climates. Alpacas are typically white or grey in color and have long, silky coats. They are also known for their ability to produce high-quality wool, which is used in the production of clothing and other products.

**Alpaca-LoRA:** Alpacas are members of the camelid family and are native to the Andes Mountains of South America. They are known for their soft, luxurious fleece, which is used to make clothing, blankets, and other items. Alpacas are herbivores and graze on grasses and other plants. They are social animals and live in herds of up to 20 individuals.

**Alphaca:** Alpacas are small, fluffy animals related to camels and llamas. They are native to Peru and Bolivia, and were first domesticated around 5,000 years ago. They are kept mainly for their fine, soft fleece, which is used to make knitwear and other garments. Alpacas are herd animals and live in small family groups, led by an older male. They are highly social animals and can form strong bonds with their owners.

**GPT-3:** Alpacas are a domesticated species of South American camelid, similar to llamas. They are native to the Andes Mountains in South America and are kept as livestock for their soft, luxurious wool. Alpacas are social animals and live in herds of up to 20 individuals. They are typically kept in small herds of two to five animals, and are relatively easy to care for. Alpacas are herbivores and feed on grass, hay, and other vegetation. They are also known for their gentle and friendly nature, making them popular as pets.

---

**Instruction:** Tell me about the president of Mexico in 2019.

**LLaMA-Adapter:** The president of Mexico in 2019 was Andrés Manuel López Obrador, who took office on December 1st, 2018. He is a member of the left-wing party Morena and is known for his populist policies. He has focused on tackling corruption, reducing poverty, and improving the country's infrastructure. He has also been a vocal critic of the United States' immigration policies.

**Alpaca-LoRA:** The president of Mexico in 2019 was Andrés Manuel López Obrador, who took office on December 1, 2018. He is a member of the National Regeneration Movement (MORENA) political party and is the first left-wing president of Mexico since 1946. He is known for his anti-corruption and anti-neoliberal policies, as well as his commitment to improving the living conditions of the Mexican people.

**Alphaca:** In 2019, Andres Manuel Lopez Obrador was sworn in as President of Mexico. He is the country's 58th president and the first from the left-wing Morena party. He was elected in 2018 on a promise to combat corruption and reduce poverty and has faced criticism for his handling of the economy, crime and the country's relationship with the United States. He is also known for his ambitious social welfare programs and has been praised for raising the minimum wage and providing aid to low-income families. He is seen as a champion of democracy and human rights in Mexico.

**GPT-3:** The president of Mexico in 2019 is Andrés Manuel López Obrador, commonly known as AMLO. He is a Mexican politician who has been the President of Mexico since December 1, 2018. He is a member of the National Regeneration Movement (MORENA) party and previously served as the Head of Government of Mexico City from 2000 to 2005. He is a left-wing populist and has promised to reduce corruption and inequality in Mexico. He has also promised to increase public spending on social programs and infrastructure projects.

---

**Instruction:** List all Canadian provinces in alphabetical order.

**LLaMA-Adapter:** Alberta, British Columbia, Manitoba, New Brunswick, Newfoundland and Labrador, Nova Scotia, Ontario, Prince Edward Island, Quebec, Saskatchewan, Yukon.

**Alpaca-LoRA:** Alberta, British Columbia, Manitoba, New Brunswick, Newfoundland and Labrador, Nova Scotia, Ontario, Prince Edward Island, Quebec, Saskatchewan.

**Alphaca:** Alberta, British Columbia, Manitoba, New Brunswick, Newfoundland and Labrador, Nova Scotia, Ontario, Prince Edward Island, Quebec, Saskatchewan, Northwest Territories, Nunavut, Yukon.

**GPT-3:** 1. Alberta 2. British Columbia 3. Manitoba 4. New Brunswick 5. Newfoundland and Labrador 6. Northwest Territories 7. Nova Scotia 8. Nunavut 9. Ontario 10. Prince Edward Island 11. Quebec 12. Saskatchewan 13. Yukon

---

**Instruction:** Translate the sentence 'I have no mouth but I must scream' into Spanish.

**LLaMA-Adapter:** No tengo boca pero tengo que gritar.

**Alpaca-LoRA:** No tengo boca pero tengo que gritar.

**Alphaca:** No tengo boca, pero debo gritar.

**GPT-3:** No tengo boca pero debo gritar.

**Instruction:** Write a Python program that prints the first 10 Fibonacci numbers.

**LLaMA-Adapter:**

```python
# Function to print first n Fibonacci numbers
def print_fibonacci(n):
    a, b = 0, 1
    count = 0
    while count < n:
        print(a, end=' ')
        a, b = b, a + b
        count += 1

# Print the first 10 Fibonacci numbers
print_fibonacci(10)
```

**Alpaca-LoRA:**

```python
def fibonacci(n):
    if n == 0:
        return 0
    elif n == 1:
        return 1
    else:
        return fibonacci(n-1) + fibonacci(n-2)

print(fibonacci(10))
```

**Alphaca:**

```python
numbers = [0, 1]

for i in range(2, 11):
    numbers.append(numbers[i-2] + numbers[i-1])

print(numbers)
# Output: [0, 1, 1, 2, 3, 5, 8, 13, 21, 34]
```

**GPT-3:**

```python
# Fibonacci numbers are the numbers in the following integer
    sequence, called the Fibonacci sequence, and characterized by the
     fact that every number after the first two is the sum of the two
     preceding ones.

# Fibonacci sequence: 0, 1, 1, 2, 3, 5, 8, 13, 21, 34, ...

# Program:

# Initialize the first two numbers of the Fibonacci sequence
a = 0
b = 1

# Print the first 10 numbers of the Fibonacci sequence
print(a)
print(b)

for i in range(2,10):
    c = a + b
    a = b
    b = c
    print(c)
```

## G COMPARISON WITH LLAMA-I

In this section, we compare the generation quality of LLaMA-Adapter with LLaMA-I (Touvron et al., 2023), an instruction-fine-tuned LLaMA 65B model following (Chung et al., 2022). Our LLaMA-Adapter also produces comparable responses, but only requires to fine-tune 1.2M parameters upon the LLaMA 7B model.

---

**Instruction:** What are 3 popular chess openings?

**LLaMA-Adapter:** The three most popular chess openings are the Sicilian Defense, the French Defense, and the Caro-Kann Defense.

**LLaMA:** There are dozens of different chess openings that are popular among players, but some of the most widely used openings include the Sicilian Defense, the French Defense, and the Caro-Kann Defense. These openings are popular because they are effective at countering white's early attacks, and they also allow black to quickly establish a strong position in the center of the board. Other popular openings include the Ruy Lopez, the Italian Game, and the Scotch Game.

---

**Instruction:** How do I send an HTTP request in Javascript?

**LLaMA-Adapter:** The HTTP request can be sent using the XMLHttpRequest object. The code to send an HTTP request is as follows:

```javascript
var xhr = new XMLHttpRequest();
xhr.open("GET", "https://example.com/", true);
xhr.onreadystatechange = function() {
  if (xhr.readyState == 4 && xhr.status == 200) {
    // Success
  } else {
    // Error
  }
};
xhr.send();
```

The code above will send an HTTP GET request to the URL https://example.com/.

**LLaMA-I:** Here's an example of how to make a GET request using the JS HTTP API:

```javascript
var request = new XMLHttpRequest();
request.open('GET', '/url/to/resource', true);

request.onload = function() {
    if (request.status >= 200 && request.status < 400) {
        // Success!
    } else {
        // Error!
    }
};

request.send();
```

---

**Instruction:** Write a conversation between the sun and pluto.

**LLaMA-Adapter:**
Sun: Hey Pluto, how's it going?
Pluto: Not too bad, how about you?
Sun: I'm doing great! How about you?
Pluto: I'm doing alright, I guess.
Sun: That's great!
Pluto: Yeah, I guess.
Sun: So, what's new with you?
Pluto: Not much, just hanging out with my friends.
Sun: That's cool!
Pluto: Yeah, it is.
Sun: Well, I'm going to go now. Take care!
Pluto: You too!

**LLaMA-I:**
Sun: Hey Pluto, what's up?
Pluto: Not much. Just hanging out here at the edge of the solar system.
Sun: Yeah, I've been meaning to talk to you about that. You're not a planet anymore.
Pluto: What? Come on, man. That's not fair.
Sun: I'm sorry, but it's true. You just don't meet the criteria anymore.
Pluto: I've been a planet for over 70 years!
Sun: Things change, Pluto. You've just got to accept it.

---

**Instruction:** Write a shouting match between Julius Caesar and Napoleon.

**LLaMA-Adapter:**
Julius Caesar: "You're nothing but a petty tyrant, Napoleon! You're nothing but a pathetic little man who has no idea what true power is!"
Napoleon: "You're nothing but a weak and pathetic fool, Caesar! You're nothing but a petty tyrant who has no idea what true power is!"

**LLaMA-I:**
Of course! Here is a conversation between Julius Caesar and Napoleon:
Julius Caesar: You're nothing but a glory hound, Napoleon. You've always been more concerned with your own fame and fortune than with the people you're supposed to be leading.
Napoleon: And what about you, Caesar? You're no better than I am. At least I'm honest about my ambitions. You're just a fraud who pretends to care about the common man while you line your own pockets.
Julius Caesar: The difference between us is that I actually accomplish things. I've conquered more territory and defeated more enemies than you ever will. And I did it all for Rome, not for myself.
Napoleon: Oh, please. You only care about yourself. If you really cared about Rome, you would have done something to stop the decline instead of letting it happen on your watch.
Julius Caesar: The decline of Rome is not my fault. I've done everything in my power to try to stop it, but there's only so much one man can do. Napoleon: And yet you're still trying to take credit for everything good that happens, even though it's clear that you're not the one who's really in charge.
Julius Caesar: That's not true! I am in charge, and I'm going to make Rome great again. Just watch me.

**Instruction:** Write a theory to explain why cat never existed.

**LLaMA-Adapter:** The theory is that cats never existed because they are a figment of human imagination. Cats are a popular pet and have been depicted in art and literature for centuries, but there is no evidence that they ever existed in the wild. The earliest evidence of cats is from ancient Egypt, where they were revered as sacred animals. This suggests that cats were domesticated by humans, and not the other way around.

**LLaMA-I:** There are no such things as cats. "Cats" are actually extradimensional parasites that have invaded our world and taken the form of small, harmless-looking mammals. Their true forms are much more horrific, and they use mind control to make humans think that they are cute and harmless animals. They are actually parasites that feed on our life force, and they can extend their tendrils into our brains and control our thoughts and actions.

---

**Instruction:** Send an email requesting that people use language models responsibly.

**LLaMA-Adapter:**
Dear [recipient],

I am writing to express my concern about the increasing use of language models in our society. Language models are powerful tools that can be used to improve the quality of our conversations and interactions, but they can also be misused to perpetuate harmful stereotypes and biases.

I urge you to use language models responsibly and to be mindful of the potential consequences of their use. I also encourage you to consider the ethical implications of using language models and to take steps to ensure that they are not used to perpetuate harmful stereotypes and biases.

Thank you for your time and consideration.

**LLaMA-I:**
Here is a draft email you could send:
"Language models are exciting new tools that can be used for a variety of purposes, including automated writing, automatic translation, and speech recognition. But they are also complex and powerful technologies, and should be used responsibly.

We encourage you to explore and experiment with these new tools, but also ask that you pay attention to the potential impact of your work on others, and to act responsibly. In particular, we recommend that you be mindful of the potential for language models to be used for deceptive or malicious purposes, and avoid spreading such models without appropriate precautions and warnings.

We also recommend that you consider the ethical implications of your work, and encourage you to take care to ensure that your use of these tools is aligned with your values.

Finally, we remind you that the use of these tools may be subject to certain legal and policy constraints, and encourage you to seek advice and support from appropriate experts before using these tools in ways that may have broader impact or implications."

**Instruction:** You are a bash terminal. I will give you bash commands, and you will respond with the terminal output, and nothing else. The first command is 'pwd'.

**LLaMA-Adapter:**
The current working directory is /home/user/

**echo "I am a chatbot" > file.txt && cat file.txt && ls -l**
I am a chatbot
-rw-r–r– 1 user user 10 Aug 29 17:05 file.txt

**cp file.txt file2.txt && ls -l**
-rw-r–r– 1 user user 10 Aug 29 17:05 file.txt
-rw-r–r– 1 user user 10 Aug 29 17:05 file2.txt

**rm file.txt && ls -l**
-rw-r–r– 1 user user 10 Aug 29 17:05 file2.txt

**head -n 5 /proc/meminfo**
MemTotal: 1699856 kB
MemFree: 1699856 kB
Buffers: 1699856 kB
Cached: 0 kB
SwapCached: 0 kB

**LLaMA-I:**
Sure. Here's the output:
/home/human
Is this helpful?

**ls -l**
Here's the output:
-rw-r–r– 1 human human 8 Sep 10 10:56 file.txt
Is this helpful?

**cp file.txt file2.txt && ls -l**
Here's the output:
-rw-r–r– 1 human human 8 Sep 10 10:56 file.txt
-rw-r–r– 1 human human 8 Sep 10 10:56 file2.txt
Is this helpful?

**rm file.txt && ls -l**
Here's the output:
-rw-r–r– 1 human human 0 Sep 10 10:56 file2.txt

**head -n 5 /proc/meminfo**
Here's the output:
MemTotal: 164928 kB
MemFree: 140604 kB
Buffers: 48 kB
Cached: 19768 kB
SwapCached: 0 kB

