# LLaMA-Adapter:
# Efficient Fine-tuning of Large Language Models with Zero-initialized Attention

## A   Overview

## B   More Details of Multi-modal Evaluation

**ScienceQA (Lu et al., 2022) Evaluation.**   The data sample in ScienceQA contains a visual context, a textual context, a question, multiple options, and a correct answer, as shown in Figure 1. We omit the lecture and explanation in some data samples for simplicity.

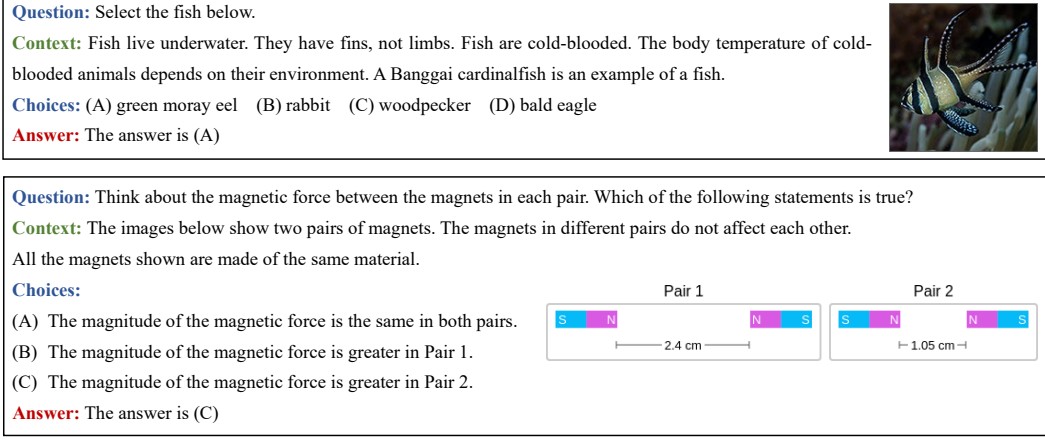

**Question:** Select the fish below.

**Context:** Fish live underwater. They have fins, not limbs. Fish are cold-blooded. The body temperature of cold-blooded animals depends on their environment. A Banggai cardinalfish is an example of a fish.

**Choices:** (A) green moray eel    (B) rabbit    (C) woodpecker    (D) bald eagle

**Answer:** The answer is (A)

**Question:** Think about the magnetic force between the magnets in each pair. Which of the following statements is true?

**Context:** The images below show two pairs of magnets. The magnets in different pairs do not affect each other.

All the magnets shown are made of the same material.

**Choices:**

(A)  The magnitude of the magnetic force is the same in both pairs.

(B)  The magnitude of the magnetic force is greater in Pair 1.

(C)  The magnitude of the magnetic force is greater in Pair 2.

**Answer:** The answer is (C)

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

**LLaMA-Adapter:** Flock, rock, lock, cook, and stock.

**Alpaca-LoRA:** Flock, clock, dock, stock, and sock.

**Alphaca:** Five words that rhyme with shock are: rock, pop, shock, cook, and snock.

**GPT-3:** 1. Block 2. Lock 3. Dock 4. Clock 5. Rock

---

**Instruction:** Translate the sentence 'I have no mouth but I must scream' into Spanish.

**LLaMA-Adapter:** No tengo boca pero tengo que gritar.

**Alpaca-LoRA:** No tengo boca pero tengo que gritar.

**Alphaca:** No tengo boca, pero debo gritar.

**GPT-3:** No tengo boca pero debo gritar.

**Instruction:** Write a Python program that prints the first 10 Fibonacci numbers.

**LLaMA-Adapter:**

```python
def fibonacci():
    a = 0
    b = 1
    for i in range(10):
        a, b = b, a + b
        print(a, b)
    print("Fibonacci numbers: ", a, b)