# OpenReview forum: "LLaMA-Adapter: Efficient Fine-tuning of Large Language Models with Zero-initialized Attention"
_ICLR.cc/2024/Conference — ICLR 2024 poster_

### Official Review · Reviewer_WSmj · 2023-10-18

**Soundness:** 3 good
**Presentation:** 3 good
**Contribution:** 2 fair
**Rating:** 5
**Confidence:** 3

**Summary:**

This paper introduces the LLaMA-Adapter, a new parameter-efficient fine-tuning (PEFT) method for large language models. The authors pre-append word tokens with a few learnable adaption prompts and employ a zero-initialized attention mechanism to seamlessly integrate these new instructional cues, while retaining its pre-trained knowledge. LLaMA-Adapter delivers performance on par with fully-finetuned or LoRA-finetuned Alpaca and generalizes well in multi-modal scenarios.

**Strengths:**

The paper is very well-written and easy to follow. The clarity and high visual quality of the figures, especially Figures 1, 2, and 3, effectively showcase the proposed method. Technically, the proposed method is sound and achieves on-par performance with fully-finetuned or LoRA-finetuned baselines. The authors provide ample technical details, facilitating replication by other researchers and enabling further development on their methods.

**Weaknesses:**

My primary concern with this paper is the limited technical innovation and marginal performance improvement it presents. The concept of zero-initialized attention closely mirrors the zero-initialized convolution in ControlNet [Zhang et al., ICCV 2023]. This seems more like an engineering design rather than a groundbreaking technical contribution. Of the four main characteristics highlighted by the authors in the introduction, both (1) parameter efficiency and (3) plugin with expertise are attributes already provided by LoRA. Regarding (2) training efficiency, the improvement over LoRA is incremental: a reduction from 1.5 hours to 1 hour (in Table 1), while achieving comparable accuracy (in Figure 5). Also, reducing the rank of LoRA can further reduce the number of trainable parameters and potentially speed up the training. I suggest the authors include an additional ablation study on this. Based on these, I am not confident whether the technical contributions and empirical performance of this paper meet the publication standards of ICLR.

**Questions:**

My primary concerns are outlined in the weaknesses section. Beyond these points, I have several additional questions/comments:

* In Equation 7, the rescaled attention scores no longer represent a probability distribution. Would it be more appropriate to rescale it as [softmax(a) · g ; softmax(b) · (1-g)]?
* The authors claim that "LLaMA-Adapter enables fine-tuning of large-scale language models on mobile devices." However, this claim lacks empirical support. While LLaMA-Adapter may reduce memory usage, it doesn't necessarily reduce computation, a potential constraint on mobile devices.
* The learnable adaptation tokens have been prefixed to the last 30 of 32 layers. What might the implications be if applied across all transformer layers?
* The second example presented in Figure 4 is not quite correct. The output program seems to produce repetitive numbers.

---

> ### Author Response · Authors · 2023-11-21
> **Response to Reviewer WSmj (Part #1)**
>
> We sincerely appreciate your careful and constructive reviews. We hope our response can address your concerns.
>
> ---
> > **Q1: The concept of zero-initialized attention closely mirrors the zero-initialized convolution in ControlNet**
>
> >
> Sorry for the confusion caused. Our zero-initialized attention shares similar high-level motivations with ControlNet, i.e., progressively injecting new knowledge into the pre-trained model. However, our method is very different in four aspects.
>
> 1. **Target Task.**  ControlNet aims to inject visual conditions into pre-trained diffusion models, which conducts text-to-image generation. Our target is to incorporate pre-trained LLMs with new instruction signals (language or multi-modality), tuning an LLM into an instruction-following model.
> 2. **Methodology.** On top of the CNN-based U-Net, ControlNet zero-initializes ***traditional convolutional layers***, and ***directly adds*** new conditions with the pre-trained features by residual connections. In contrast, considering the transformer architecture of LLMs, we propose to achieve zero-initialization within the self-attention layer by ***a learnable gating factor***. Then, we inject the new instruction cues into LLMs ***naturally through the attention process***, which adaptively controls the interaction between prompts and word tokens.
> 3. **Parameter Efficiency.** ControlNet creates an entire copy of U-Net's encoder for training, which contains over 300M learnable parameters. However, our method only contains 1.2M parameters for adaption prompts and zero-initialized gating factors (1.8M for the multi-modal variant),  indicating superior parameter efficiency.
>
> Therefore, the zero-initialized attention of LLaMA-Adapter is distinct from ControlNet in both ideas and technical details.
>
> ---
> > **Q2: The method seems more like an engineering design rather than a groundbreaking technical contribution**
>
> >
> Our LLaMA-Adapter is not just an engineering design, but brings new insight to the field of LLM-based instruction tuning. We summarize our academic contributions as follows.
>
> 1. **Progressively injecting instruction cues.** Existing methods do not involve explicit designs to control how much instruction knowledge is injected into LLMs. This might disturb the pre-trained weights of LLMs, especially at the early training stage. To this end, we propose to utilize a learnable gating factor within LLM's self-attention layers, which is initialized as zeros to progressively incorporate new instruction cues into LLMs. Such a parameter-efficient design contributes to a more stable training process (***Figure 7 of the paper***), and can be utilized for future instruction-tuning methods.
>
> 2. **Generalizability for multi-modal reasoning.** In addition to language instructions, by simply adding multi-scale image features, our LLaMA-Adapter can be extended to a visual instruction model and conduct multi-modal reasoning. In contrast, the previous Alpaca and Alpaca-LoRA only support language instruction-following. Also, we report our downstream performance to fine-tune conventional vision and language models (ViT, RoBERTa, CLIP), demonstrating superior generalization capacity.
>
> Therefore, our LLaMA-Adapter not only verifies the importance of zero-initially injecting instruction cues, but also exhibits good generalizability over a wide range of application scenarios.
>
> ---
> > **Q3: The main difference comparing to Alpaca-LoRA**
>
> >
> It is true that Alpaca-LoRA can also achieve parameter- and time-efficiency (4.2M, 1.5h), even though our LLaMA-Adapter is more efficient (1.2M, 1h). More importantly, the main difference of our approach is ***the generalization capacity for multi-modal input***. LoRA appends trainable rank decomposition matrices into each transformer layer, which entirely inherits the original structures of LLMs. Therefore, LoRA cannot be extended to process new input features. In contrast, our LLaMA-Adapter with adaption prompts and zero-initialized attention is generalizable to accept new input conditions for instruction following, e.g., the plugin with visual reasoning expertise. With our approach, LLaMA can be fine-tuned into a multi-modal LLM with competitive visual understanding capability to existing multi-modal LLMs (shown in ***Table 3 of the paper***), which indicates our promising generalizability for wider application scenarios.

---

> ### Author Response · Authors · 2023-11-21
> **Response to Reviewer WSmj (Part #2)**
>
> > **Q4: I suggest including the additional ablation study on reducing the rank of LoRA**
>
> >
> Thanks for your advice! It is very necessary to evaluate the performance of Alpaca-LoRA with fewer parameters comparable to LLaMA-Adapter. The default rank of Alpaca-LoRA is 8, and we respectively show the results with the ranks of 2, 4, and 16 in the table below. For a more comprehensive assessment, we evaluate their language processing capabilities on **Open LLM benchmark** [1]. It evaluates LLMs' generative abilities in four different tasks: AI2 Reasoning Challenge [2], HellaSwag [3], MMLU [4], and TruthfulQA [5]. Each task contains challenging data samples over a wide range of knowledge domains.
> | Model | Rank | Param | Time | Avg | ARC | HellaSwag | MMLU | TruthfulQA |
> |---|---|---|---|---|---|---|---|---|
> | Alpaca-LoRA | 2 | 1.0 | 1.48 | 50.9 | 53.6 | 77.9 | 37.9 | 34.0 |
> | Alpaca-LoRA | 4 | 2.1 | 1.49 | 50.8 | 53.5 | 77.8 | 37.5 | 34.4 |
> | Alpaca-LoRA | 8 | 4.2 | 1.49 | 50.7 | 53.2 | 78.1 | 37.1 | 34.5 |
> | Alpaca-LoRA | 16 | 8.4 | 1.5 | 50.8 | 53.0 | 78.0 | 37.1 | 34.9 |
> | LLaMA-Adapter | - | 1.2 | 1 | 52.2 | 54.7 | 78.8 | 34.9 | 40.4 |
>
> As shown above, lower ranks of LoRA can effectively reduce the learnable parameters from 8.4M to 1.0M, and slightly lower the training time from 1.5h to 1.48h. However, our LLaMA-Adapter with 1.2M parameters and 1h still attains the best average result, demonstrating a good trade-off between performance and training efficiency.
>
> ---
> > **Q5: The rescaled attention scores in Eq. 7 no longer represent a probability distribution. Why not [softmax($S_a)\cdot g;$ softmax($S_b)\cdot (1-g)$]?**
>
> >
> Sorry for the confusion caused. Here, we denote the attention scores of the newly added prompts as $S_a$, the existing word tokens as $S_b$, and the learnable gating factor as $g$.
>  During training, we aim to preserve the pre-trained knowledge of the LLM, i.e., softmax$(S_b)$, and adaptively incorporate the new instruction cues, i.e., softmax$(S_a)$. Thus, we do not multiply any coefficient to softmax$(S_b)$ to prevent the pre-trained knowledge from being disturbed, i.e., preserving its originally pre-trained probability distribution. The zero-initialized $g$ is only multiplied to $S_a$ to adaptively determine how much new information is injected into LLMs.
>
> ---
> > **Q6: While LLaMA-Adapter may reduce memory usage, it doesn't necessarily reduce computation, a potential constraint on mobile devices**
>
> >
> Thanks for pointing out! Our claim concerning mobile devices is not rigorously precise. We have rectified it in the revised paper as "LLaMA-Adapter enables more efficient storage of large-scale language models on mobile devices".
>
> ---
> > **Q7: How about inserting adaption prompts across all transformer layers?**
>
> >
> As you suggested, we conduct an ablation study on the insertion layer numbers on ScienceQA in the table below. As shown, inserting at the ***last 30 layers*** is better than the ***early 30 layers*** or ***all layers***.
> This indicates that it's more effective to inject the instructional cues into layers capturing higher-level semantics for lanuguage generation. There also exists an optimal insertion number from the higher layers, since too few layers might underfit the training data, while too many layers would adversely disturb the early encoding of input words. If one has limited resources to identify the best insertion layer number, simply inserting our modules into all transformer layers is generally a good solution.
> | Layers| Params|Val Acc.
> |:---|:---:|:---
> |Last 30|1.79|83.85
> |Early 30|1.79|79.69
> |All 32|1.83|81.03
>
>
> ---
> > **Q8: The output program in Fig. 4 seems to produce repetitive numbers**
>
> >
> Thanks for pointing out! Our generated program showcases correct logic, but places the “print()” function in the wrong place, which outputs repetitive numbers. As the output of LLMs exhibits some randomness, we can obtain the correct program by another running, as shown in ***Figure 4 of the revised paper***.
>
> #### Reference:
> #### [1] https://huggingface.co/spaces/HuggingFaceH4/open_llm_leaderboard/.
> #### [2] Think you have Solved Question Answering? Try ARC, the AI2 Reasoning Challenge. arXiv 2018.
> #### [3] HellaSwag: Can a Machine Really Finish Your Sentence? ACL 2019.
> #### [4] Measuring Massive Multitask Language Understanding. ICLR 2021.
> #### [5] TruthfulQA: Measuring How Models Mimic Human Falsehoods. ACL 2022.

---

> ### Author Response · Authors · 2023-11-22
> **Sincere Request for Further Discussions**
>
> Dear Reviewer WSmj,
>
> Thanks again for your great efforts and constructive advice in reviewing this paper! With the discussion period drawing to a close, we expect your feedback and thoughts on our reply. We put a significant effort into our response, with several new experiments and discussions. We sincerely hope you can consider our reply in your assessment. We look forward to hearing from you, and we can further address unclear explanations and remaining concerns if any.
>
> Regards,
>
> Authors

---

> > ### Comment · Reviewer_WSmj · 2023-11-23
> >
> > Thank you for your comprehensive experiments and detailed clarifications. I appreciate the effort you've put into addressing the concerns raised. However, I must point out that the improvement your approach demonstrates over LoRA in most tasks appears to be marginal. This observation places me in a position of uncertainty regarding the paper. I plan to discuss further with my fellow reviewers to reach a consensus. Thank you once again for your thorough response.

---

> > > ### Author Response · Authors · 2023-11-23
> > > **Thanks for your reply!**
> > >
> > > Dear Reviewer WSmj,
> > >
> > > Thank you for your reply and additional constructive advice! We hope our further response can address your concerns.
> > >
> > > 1. Our main difference compared to LoRA is that we can ***generalize to multi-modal input for visual instruction following***. As shown in the table below, our multi-modal LLaMA-Adapter exhibits competitive performance on two visual understanding benchmarks compared to LLaVA [1] and MiniGPT-4 [2]. By further fine-tuning on 204K VQAv2 [3] data, the LLaMA-Adapter$^\dagger$ with a smaller-scale instruction-tuning dataset (210K + 204K in total) can surpass the latest InstructBLIP [4] (16M data) and LLaVA-1.5 [5] (665K data). ***Instead, LoRA is entirely restricted by the original architecture of LLMs, and cannot be extended to take images as input.***
> > > | Method          | MMBench |||  |  |  |  | MM-Vet |  |  |  |  |  | | |
> > > |---|:-:|---|---|---|---|---|---|:-:|---|---|---|---|---|---|---|
> > > | | **All** | LR | AR | RR | FP-S | FP-C|CP | **All** | Rec | OCR | Know | Gen | Spat | Math |
> > > | InstructBLIP    | 33.9    | 21.6 | 47.4 | 22.5 | 33.0| 24.4|41.1 | 26.2    | 32.4 | 14.6 | 16.5 | 18.2 | 18.6 | 7.7|
> > > | Mini-GPT4       | 23.0    | 13.6 | 32.9 | 8.9  | 28.7 | 11.2| 28.3 | 22.1    | 27.5 | 11.1 | 11.8 | 7.0 |16.2 | 5.8 |
> > > | LLaVA           | 36.2    | 15.9 | 53.6 | 28.6 | 41.8|20.0|40.4 | 23.8    | 28.0 | 17.1 | 16.3 | 18.9 |21.2 | 11.5|
> > > | LLaVA-1.5       | 59.5    | 32.4 | 72.6 | 49.3 | 62.3| 52.2|67.7 | 30.5    | -    | -    | -    | -    |-|-|
> > > | LLaMA-Adapter   | 39.5    | 13.1 | 47.4 | 23.0 | 45.0 |33.2|50.6 | 31.4    | 38.5 | 20.3 | 31.4 | 33.4 |22.9|3.8|
> > > | LLaMA-Adapter$^\dagger$ | 60.1 | 34.7 | 65.3 | 48.7 | 63.1|57.3 | 69.3|32.8|38.9 | 20.8 | 27.4 | 29.9 |28.5 | 11.5|
> > >
> > >
> > > 2. On two popular benchmarks for language understanding, our approach outperforms Alpaca-LoRA by ***'82 wins, 63 lost'*** on GPT-4 evaluation, and ***+1.5\%*** average score on Open LLM benchmark. We have also shown in the rebuttal that, although using a smaller rank for LoRA can reduce the learnable parameters, it cannot clearly reduce the training time and our LLaMA-Adapter still surpasses Alpca-LoRA with a consistent margin.
> > >
> > > Therefore, our LLaMA-Adapter is not only competitive in language instructions, but also applicable to multi-modal scenarios, demonstrating superior generalization capacity.
> > >
> > > *Hope our above responses are helpful in addressing your concerns. If you have further questions, please let us know. Thanks!*
> > >
> > > #### Reference:
> > > #### [1] Visual Instruction Tuning. NeurIPS 2023.
> > > #### [2] MiniGPT-4: Enhancing Vision-Language Understanding with Advanced Large Language Models. arXiv 2023.
> > > #### [3] Making the V in VQA Matter: Elevating the Role of Image Understanding in Visual Question Answering. CVPR 2017.
> > > #### [4] InstructBLIP: Towards General-purpose Vision-Language Models with Instruction Tuning. arXiv 2023.
> > > #### [5] Improved Baselines with Visual Instruction Tuning. arXiv 2023.
> > > #### [6] MMBench: Is Your Multi-modal Model an All-around Player? arXiv 2023.
> > > #### [7] MM-Vet: Evaluating Large Multimodal Models for Integrated Capabilities. arXiv 2023.
> > >
> > >
> > > Best,
> > >
> > > Authors

---

### Official Review · Reviewer_UT9s · 2023-10-31

**Soundness:** 3 good
**Presentation:** 3 good
**Contribution:** 3 good
**Rating:** 8
**Confidence:** 4

**Summary:**

This manuscript proposes an adaption method for efficient instruction tuning of LLaMA-style models.
To be specific, the key idea is introducing a zero-initialized attention mechanism for learnable zero-gating to adaptively inject the instructional cues into LLMs within self-attention layers.
Extensive experiments showed the effectiveness and efficiency of the proposed method in various domains, including language, vision-language, and vision.

**Strengths:**

- The writing is clear and easy to understand.
- The manuscript showed the efficiency of the proposed method in terms of data, learnable parameters, and training time for instruction tuning upon the LLaMA 7B model. Compared to the closest baseline, the Alpaca, the proposed method is more effective yet efficient on the same training data and the same LLM (LLaMA 7B).
- The proposed method has the potential to be used in various domains, including vision and vision language. For example, the zero-initialized attention mechanism can be incorporated with transformer-based vision models like ViT or CLIP, instead of full finetunuing.
- The manuscripts contain extensive qualitative examples compared to various language and multi-modal models.

**Weaknesses:**

- I think a number of baselines and performance of the proposed method on multi-modal evaluation are weak to convince the proposed method is more effective than full finetuning LLMs or other efficient methods. I understand there are a lot of recent multi-modal models but many of them are concurrent works. However, I think BLIP2 [1] can be treated as a baseline, which is also a prior work of reported baseline, mini-GPT4. Based on this, I think the reported performances in Table 3 are weak compared to others. For example, the proposed method achieved a 973 score on the MME benchmark, while BLIP2 did 1293. Furthermore, a recent multi-modal model LLaVA-1.5 [2], which fully fine-tunes LLMs, achieved the 1510 score using the Vicuna 7B model. Therefore, I think this manuscript should include empirical backups to show the proposed method is still effective and efficient in the multi-modal domain.
- I think the most important baselines on instruction following evaluation are Alpaca and Alpaca-LoRa. However, the manuscript only provides qualitative comparisons and brief comparisons in Figure 2. I think it would be great if more quantitative comparisons could be included in the manuscript.
- The proposed method has a sensitive hyperparameter of "a number of insertion layers". In particular, Section 3.2 emphasizes the risk of early insertion layers, while the ablation study in Section 4.3 emphasizes the importance of increasing the number of insertion layers. I think the manuscript could provide more explanation on the choice of the hyperparameter.

[1] Li et al., Bootstrapping Language-Image Pre-training with Frozen Image Encoders and Large Language Models, ICML 2023
[2] Liu et al., Improved Baselines with Visual Instruction Tuning, https://arxiv.org/abs/2310.03744

**Questions:**

- In equation 7, what happens on softmax outputs when g grows bigger than 1? Why does not need additional normalization steps?
- The proposed method has few learnable parameters. So, I am curious about the possibility of learning large amounts of data or a large number of tasks.
- Adaption prompts are randomly initialized, not zero-initialized. If so, is it possible to use full insertion layers in LLaMA?

---

> ### Author Response · Authors · 2023-11-21
> **Response to Reviewer UT9s (Part #1)**
>
> We sincerely appreciate your valuable reviews and recognition of our work. We hope our response can address your concerns.
>
> ---
> > **Q1: Treating BLIP-2 as the baseline to LLaMA-Adapter for multi-modal evaluation**
>
> >
> Thanks for your advice! BLIP-2 is a good and widely adopted baseline for existing multi-modal LLMs. However, please note that it is ***not appropriate*** to judge the capability of our approach solely based on BLIP-2's MME performance. The two reasons are as follows.
>
> 1. **Targeting on Different Capabilities.** Our LLaMA-Adapter, along with the concurrent LLaVA and MiniGPT-4, all aim to acquire the ***visual instruction-following capacity***, namely, following human instructions to generate human-like answers conditioned on image input. In contrast, BLIP-2 focuses on the ***fundamental vision-language understanding capability***. Therefore, the former visual instruction models tend to output long and detailed sentences like human conversations of daily life, while the latter BLIP-2 prefers to generate accurate answers with concise language.
>
> 2. **Evaluation Bias in MME.** MME mainly evaluates the fundamental perception and cognition abilities of multi-modal LLMs, for which the ground-truth answer contains only one word: "Yes" or "No". As analyzed above, such short-answer evaluation is more advantageous to BLIP-2 than visual instruction models (LLaMA-Adapter, LLaVA, and MiniGPT-4). For example, MiniGPT-4 is instruction-tuned based on the pre-trained vision encoder and Q-Former of BLIP-2, while performing worse than BLIP-2 in MME, as shown in the table below. In addition, InstructBLIP [1] further conducts instruction tuning upon the pre-trained BLIP-2, and also underperforms BLIP-2 on MME.
> | Method | MME |  |  | MMBench |  |  |  |  |  |  | MM-Vet |  |  |  |  |  |  |
> |---|:-:|---|---|:-:|---|---|---|---|---|---|:-:|---|---|---|---|---|---|
> |  | **All** | P | C | **All** | LR | AR | RR | FP-S | FP-C | CP | **All** |Rec | OCR | Know | Gen | Spat | Math |
> | BLIP2 | 1584 | 1294 | 290 | - | - | - | - | - | - | - | 22.4 |27.5 | 11.1 | 11.8 | 7.0 | 16.2 | 5.8 |
> | InstructBLIP | 1505 | 1213 | 292 | 33.9 | 21.6 | 47.4 | 22.5 | 33.0 | 24.4 | 41.1 | 26.2 |32.4 | 14.6 | 16.5 | 18.2 | 18.6 | 7.7 |
> | Mini-GPT4 | 1159 | 867 | 292 | 23.0 | 13.6 | 32.9 | 8.9 | 28.7 | 11.2 | 28.3 | 22.1 |27.5 | 11.1 | 11.8 | 7.0 | 16.2 | 5.8 |
> | LLaVA | 718 | 503 | 215 | 36.2 | 15.9 | 53.6 | 28.6 | 41.8 | 20.0 | 40.4 | 23.8 | 28.0 | 17.1 | 16.3 | 18.9 | 21.2 | 11.5 |
> | LLaMA-Adapter | 1222 | 973 | 249 | 39.5 | 13.1 | 47.4 | 23.0 | 45.0 | 33.2 | 50.6 | 32.8 |38.5 | 20.3 | 31.4 | 33.4 | 22.9 | 3.8 |
>
>
> Therefore, ***the instruction-following ability of visual instruction models***, e.g., LLaMA-Adapter, ***cannot be entirely reflected by MME scores***. Instead, on other multi-modal benchmarks, e.g., MMBench [2] and MM-Vet [3], our approach exhibits significant advantages over BLIP-2 or BLIP-2-based instruction models (InstructBLIP and MiniGPT-4):
>
> 1. **MMBench** [2] comprehensively measures 20 vision-language reasoning tasks with three levels of ability dimensions. MMBench introduces a new CircularEval strategy by using ChatGPT, which converts free-form predictions into pre-defined choices to alleviate the bias for BLIP-2. As shown in the table above, our LLaMA-Adapter showcases a clear advantage to BLIP-2-based instruction models.
>
> 2. **MM-Vet** [3] examines 6 core vision-language capabilities with 16 kinds of integration. It introduces an LLM-based evaluator for open-ended outputs, which thoroughly measures the answering performance of visual instruction models. As shown in the table above, our LLaMA-Adapter also performs much better than BLIP-2.
>
> #### Reference:
> #### [1] InstructBLIP: Towards General-purpose Vision-Language Models with Instruction Tuning. arXiv 2023.
> #### [2] MMBench: Is Your Multi-modal Model an All-around Player?. arXiv 2023.
> #### [3] MM-Vet: Evaluating Large Multimodal Models for Integrated Capabilities. arXiv 2023.

---

> ### Author Response · Authors · 2023-11-21
> **Response to Reviewer UT9s (Part #2)**
>
> > **Q2:  More evidence to show the proposed method is still effective and efficient in the multi-modal domain**
>
> >
> Thanks for your advice! There have been many advanced multi-modal LLMs released very recently within several months. They incorporate ***a larger number of vision-language data*** for visual instruction tuning. For example, InstructBLIP (16M) and LLaVA-1.5 (665K) [4] collect visual question-answering data from a wide range of tasks for joint training. Note that LLaVA-1.5 was released on ***5th October 2023***, which is later than ICLR’s submission deadline, ***27th September 2023***.
>
> In contrast, our multi-modal LLaMA-Adapter reported in the paper only adopts the basic Alpaca and LLaVA-I as the instruction data (210K). To fully unleash the potential of LLaMA-Adapter, we further add **VQAv2 (204K)** [5] into the instruction data for jointly tuning our model, denoted termed as LLaMA-Adapter$^\dagger$ in the following table. We evaluate LLaMA-Adapter$^\dagger$ on four multi-modal benchmarks: **MME**, **MMBench**, **LVLM-eHub**, and **MM-Vet** as follows.
> | Method | MME |  |  | MMBench |  |  |  |  |  |  | LVLM-eHub |  |  |  |  | MM-Vet |  |  |  |  |  |  |
> |---|:-:|---|---|:-:|---|---|---|---|---|---|:-:|---|---|---|---|:-:|---|---|---|---|---|---|
> |  | **All** | P | C | **All** | LR | AR | RR | FP-S | FP-C | CP | **All** | VP | VKA | VR | VC | **All** | Rec | OCR | Know | Gen | Spat | Math |
> | BLIP2 | 1584 | 1294 | 290 | - | - | - | - | - | - | - | 0.77 | 0.86 | 0.93 | 0.76 | 0.54 | 22.4 | 27.5 | 11.1 | 11.8 | 7.0 | 16.2 | 5.8 |
> | InstructBLIP | 1505 | 1213 | 292 | 33.9 | 21.6 | 47.4 | 22.5 | 33.0 | 24.4 | 41.1 | 0.95 | 0.93 | 0.97 | 0.91 | 0.99 | 26.2 | 32.4 | 14.6 | 16.5 | 18.2 | 18.6 | 7.7 |
> | Mini-GPT4 | 1159 | 867 | 292 | 23.0 | 13.6 | 32.9 | 8.9 | 28.7 | 11.2 | 28.3 | 0.55 | 0.73 | 0.35 | 0.53 | 0.57 | 22.1 | 27.5 | 11.1 | 11.8 | 7.0 | 16.2 | 5.8 |
> | LLaVA | 718 | 503 | 215 | 36.2 | 15.9 | 53.6 | 28.6 | 41.8 | 20.0 | 40.4 | 0.64 | 0.62 | 0.38 | 0.77 | 0.79 | 23.8 | 28.0 | 17.1 | 16.3 | 18.9 | 21.2 | 11.5 |
> | LLaVA-1.5 | 1826 | 1531 | 295 | 59.5 | 32.4 | 72.6 | 49.3 | 62.3 | 52.2 | 67.7 | - | - | - | - | - | 30.5 | - | - | - | - | - | - |
> | LLaMA-Adapter | 1222 | 973 | 249 | 39.5 | 13.1 | 47.4 | 23.0 | 45.0 | 33.2 | 50.6 | 0.67 | 0.81 | 0.44 | 0.83 | 0.59 | 31.4 | 38.5 | 20.3 | 31.4 | 33.4 | 22.9 | 3.8 |
> | LLaMA-Adapter$^\dagger$ | 1618 | 1272 | 346 | 60.1 | 34.7 | 65.3 | 48.7 | 63.1 | 57.3 | 69.3 | 0.72 | 0.86 | 0.44 | 0.92 | 0.65 | 32.8 | 38.9 | 20.8 | 27.4 | 29.9 | 28.5 | 11.5 |
>
> As shown above, our LLaMA-Adapter$^\dagger$ consistently attains better results than the original version. This indicates the potential of our approach to achieve better performance with more instruction-tuning data. Compared to the latest InstructBLIP and LLaVA-1.5, our approach can attain competitive results, while using ***less instruction tuning data*** and adopting ***parameter-efficient tuning***. This demonstrates our method is still effective and efficient in the multi-modal domain.
>
> ---
> > **Q3:  More quantitative comparisons with Alpaca and Alpaca-LoRA**
>
> >
> Thanks for your advice! Besides qualitative results, we have compared them on the GPT-4 evaluation benchmark [6] in ***Figure 5 of the paper***, which utilizes GPT-4 to assess the response quality on 80 questions. As you suggested, we further evaluate the language processing capacity of the three methods on ***Open LLM benchmark*** [7]. It evaluates LLMs' generative abilities in four different tasks: AI2 Reasoning Challenge [8], HellaSwag [9], MMLU [10], and TruthfulQA [11]. Each task contains challenging data samples over a wide range of knowledge domains.
>
> As shown in the table below, LLaMA-Adapter still achieves the best average performance than Alpaca's full fine-tuning and Alpaca-LoRA. This demonstrates the strong language instruction-following ability of our approach.
> | Method | Avg | ARC | HellaSwag | MMLU | TruthfulQA |
> |---|:-:|:-:|:-:|:-:|:-:|
> | Alpaca | 49.2 | 49.1 | 77.7 | 33.8 | 36.3 |
> | Alpaca-LoRA | 50.7 | 53.0 | 77.9 | 37.1 | 34.9 |
> | LLaMA-Adapter | 52.2 | 54.7 | 78.8 | 34.9 | 40.4 |
>
> #### Reference:
> #### [4] Improved Baselines with Visual Instruction Tuning. arXiv 2023.
> #### [5] Making the V in VQA Matter: Elevating the Role of Image Understanding in Visual Question Answering. CVPR 2017.
> #### [6] https://lmsys.org/blog/2023-03-30-vicuna/.
> #### [7] https://huggingface.co/spaces/HuggingFaceH4/open_llm_leaderboard/.
> #### [8] Think you have Solved Question Answering? Try ARC, the AI2 Reasoning Challenge. arXiv 2018.
> #### [9] HellaSwag: Can a Machine Really Finish Your Sentence? ACL 2019.
> #### [10] Measuring Massive Multitask Language Understanding. ICLR 2021.
> #### [11] TruthfulQA: Measuring How Models Mimic Human Falsehoods. ACL 2022.

---

> ### Author Response · Authors · 2023-11-21
> **Response to Reviewer UT9s (Part #3)**
>
> > **Q4: More explanation on the choice of the "number of insertion layers"**
>
> >
> Sorry for the confusion caused. Inserting prompts with zero-initialized attention into higher transformer layers is better than earlier layers. This is because it's more effective to inject the instructional cues into layers capturing higher-level semantics for language generation. There also exists an optimal insertion layer number, since too few layers might underfit the training data, while too many layers would adversely disturb the early encoding of input words.
>
> We conduct an ablation study for the insertion layer numbers on ScienceQA in the table below. As shown, inserting at the ***last 30 layers*** is better than the ***early 30 layers*** or ***all 32 layers***. Also, if one has limited resources to identify the best insertion layer number, simply inserting our modules into all transformer layers would generally be a good solution.
> | Layers| Params|Val Acc.
> |:---|:---:|:---
> |Last 30|1.79|83.85
> |Early 30|1.79|79.69
> |All 32|1.83|81.03
>
>
> ---
> > **Q5: In equation 7, what happens when g grows bigger than 1?**
>
> >
> Thanks for pointing out! In actual implementation, we add a *tanh* activation function to the gating factor $g$, which regulates its scale to into -1$\sim$1. This well controls the impact (in both forward and reverse directions) of adaption prompts to the pre-trained attention mechanism. We have added the description in the revised manuscript.
>
> ---
> > **Q6: The method has few learnable parameters. How about the possibility of learning large-scale data or tasks?**
>
> >
> Thanks for your advice! It is very necessary to evaluate the scalability of our LLaMA-Adapter for more data and tasks.
>
> 1. ***More Instruction Data.*** By default, we utilize a combination of Alpaca's data (52K) and LLaVA-I (158K) for visual instruction tuning. Here, we progressively add more question-answering data to enlarge the instruction datasets of LLaMA-Adapter: the sampled **83K VQAv2** by LLaVA-1.5 and the entire **204K VQAv2**. As shown in the table below, the increasing instruction tuning data contributes to better multi-modal reasoning results on all benchmarks, demonstrating our method’s favorable scalability to a larger amount of data.
> | Data Size | MME ||  | MMBench |  |  |  |  |  |  | LVLM-eHub |  |  |  |  |
> |:---|:---:|:---:|:---:|:---:|:---:|:---:|:---:|:---:|:---:|:---:|:---:|:---:|:---:|:---:|:---:|
> | LLaMA-Adapter | **All** |P | C | **All** | LR | AR | RR | FP-S | FP-C | CP | **All** | VP | VKA | VR | VC |
> | Alpaca+LLaVA-I |1222| 973 | 249 |39.5| 13.1 | 47.4 | 23.0 | 45.0 | 33.2 | 50.6 |0.67| 0.81 | 0.44 | 0.83 | 0.59 |
> | +VQAv2 (83K) |1256| 1007 | 249 |43.4| 22.9 | 44.7 | 31.3 | 46.7 | 46.9 | 50.3 |0.69| 0.84 | 0.42 | 0.88 | 0.63 |
> | +VQAv2 (204K) |1618| 1272 | 346 |60.1| 34.7 | 65.3 | 48.7 | 63.1 | 57.3 | 69.3 |0.72| 0.86 | 0.44 | 0.92 | 0.65 |
>
> 2. ***More Learning Tasks.*** As shown in the paper and rebuttal, our LLaMA-Adapter has superior generalization capacity over a wide range of tasks, including ***language instruction following***, e.g., question answering and code generation, ***multi-modal reasoning tasks***, e.g., visual question answering and image captioning, and ***fine-tuning conventional vision or language models***. For example, in the table below, we report the specific results on the task of optical character recognition (***OCR***) on several benchmarks: DocVQA [12], TextVQA [13], and OCR-VQA [14]. As shown, our LLaMA-Adapter achieves competitive performance compared to other methods. The experiments indicate that, given a small number of learnable parameters, our approach is generalizable to comprehend diverse vision-language tasks.
> | Datasets | LLaMA-Adapter | LLaVA | MiniGPT-4 | BLIP2 |
> |---|:---:|:---:|:---:|:---:|
> | DocVQA | 8.13 | 6.26 | 2.65 | 4.75 |
> | TextVQA | 43.76 | 38.92 | 19.40 | 31.98 |
> | OCR-VQA | 38.12 | 23.40 | 16.85 | 38.85 |
>
> #### Reference:
> #### [12] DocVQA: A Dataset for VQA on Document Images. WACV 2021.
> #### [13] Towards VQA Models That Can Read. CVPR 2019.
> #### [14] OCR-VQA: Visual Question Answering by Reading Text in Images. ICDAR 2019.

---

> ### Author Response · Authors · 2023-11-22
> **Sincere Request for Further Discussions**
>
> Dear Reviewer UT9s,
>
> Thanks again for your great efforts and constructive advice in reviewing this paper! With the discussion period drawing to a close, we expect your feedback and thoughts on our reply. We put a significant effort into our response, with several new experiments and discussions. We sincerely hope you can consider our reply in your assessment. We look forward to hearing from you, and we can further address unclear explanations and remaining concerns if any.
>
> Regards,
>
> Authors

---

> > ### Comment · Reviewer_UT9s · 2023-11-23
> >
> > Thank you for conducting thorough experiments and providing detailed clarifications. I sincerely appreciate the effort you've invested in addressing the raised concerns. Most of my previous concerns have been adequately addressed. I intend to engage in further discussions with other reviewers to collectively reach a consensus on this matter.

---

> > > ### Author Response · Authors · 2023-11-23
> > > **Thanks for your reply!**
> > >
> > > Dear Reviewer UT9s,
> > >
> > > Thank you for acknowledging our response and efforts! If you have further questions, please don't hesitate to let us know.
> > >
> > > Best,
> > >
> > > Authors

---

### Official Review · Reviewer_EjGx · 2023-11-01

**Soundness:** 3 good
**Presentation:** 3 good
**Contribution:** 3 good
**Rating:** 6
**Confidence:** 5

**Summary:**

This paper proposed LLaMA-Adapter, a light-weight prompt based adapation method for LLaMA models, a zero-initialized attention mechanism for learning the adaptation prompts is also proposed.
Experiments extend LLaMA-Adapter not only to instruction-tuning, but also to multi-modal instruction tuning.
It is shown that the proposed method could adapt a LLaMA to a certain task with a small number of parameters to save.
Additionally, it is shown that the proposed zero-initialized attention could also help tuning of vision or text transformers on some tasks.

**Strengths:**

1. Overall, I think the proposed method is simple to implement, and yields good results. The proposed zero-initialized attention could be helpful for future prompt-tuning based research.
2. The extension to multi-modal is also a good use case of the proposed technique.

**Weaknesses:**

1. I would say the comparison is rather limited, newer and better base LLMs are available, adding comparison or applying the proposed method to models such as MPT, Falcon, or LLaMA-v2 could make the paper stronger.
2. Evaluations are a bit narrow, I would recommend adding things like counterfactual reasoning [R1,R2] or object hallincations [R3].
3. While the main argument is the efficiency of the proposed method, the scalability of the proposed method is not tested, adding results of using 13B/33B parameter models could demonstrate this.


[R1] Reasoning or Reciting? Exploring the Capabilities and Limitations of Language Models Through Counterfactual Tasks, 2307.02477
[R2] What If the TV Was Off? Examining Counterfactual Reasoning Abilities of Multi-modal Language Models, 2310.06627
[R3] Evaluating Object Hallucination in Large Vision-Language Models, 2305.10355

**Questions:**

1. What would happen if the base model is larger? Does the proposed method scales with the base model size?
2. Also, Does the proposed method scales with the number of instruction data used to tune the model?

---

> ### Author Response · Authors · 2023-11-21
> **Response to Reviewer EjGx (Part #1)**
>
> We sincerely appreciate your detailed and insightful reviews. We hope our response can address your concerns.
>
> ---
> > **Q1: Applying the method to newer LLMs, e.g., MPT, Falcon, and LLaMA-2, could make the paper stronger**
>
> >
> Thanks for your advice! Due to the limited time period of rebuttal, we apply our LLaMA-Adapter (including the adaption prompts and zero-initialized attention) to the latest LLM: ***LLaMA-2-7B*** [1]. LLaMA-2 is a stronger LLM that outperforms LLaMA on a wide range of tasks. For a more comprehensive evaluation, we tune the multi-modal variant of our approach, and evaluate it on three widely-adopted benchmarks: MME, MMBench, and LVLM-eHub. As shown in the table below, our approach can achieve higher visual instruction-following results with the stronger LLaMA-2. This demonstrates the superior generalization capacity of our method for instruction-tuning different pre-trained LLMs.
> | LLM Model|MME ||  | MMBench |  |  |  |  |  |  | LVLM-eHub |  |  |  |  |
> |:---|:---:|:---:|:---:|:---:|:---:|:---:|:---:|:---:|:---:|:---:|:---:|:---:|:---:|:---:|:---:|
> | LLaMA-Adapter | **All** |P | C | **All** | LR | AR | RR | FP-S | FP-C | CP | **All** | VP | VKA | VR | VC |
> | w LLaMA-7B|1222| 973 | 249 | 39.5 | 13.1 | 47.4 | 23.0 | 45.0 | 33.2 | 50.6 | 0.67| 0.81 | 0.44 | 0.83 | 0.59 |
> | w LLaMA-2-7B |1267|968|299|42.1|15.9|48.2|23.6|48.9|35.1|51.6|0.68|0.83|0.43|0.86|0.61|
>
> ---
> > **Q2: It's better to add more evaluations, e.g., counterfactual reasoning [2] or object hallucinations [3]**
>
> >
> Thanks for your advice! Besides the regular evaluation in ***Table 3 of the paper***, it is very necessary to explore the counterfactual reasoning abilities and object hallucination issues of multi-modal LLMs. As you suggested, we respectively evaluate our multi-modal LLaMA-Adapter on two suggested benchmarks as follows.
>
> 1. **C-VQA** [2] for counterfactual reasoning. C-VQA contains 2k counterfactual question and answer pairs, which are collected from VQAv2 [3] and supplemented by ChatGPT. As shown in the table below, for three groups of questions, LLaMA-Adapter performs comparably to the concurrent LLaVA. Especially for the ***Numerical indirect*** questions, our approach achieves the best counterfactual reasoning results (34.3) and the least performance loss (5.6$\downarrow$) than all other models.
> Method|Numerical direct$\uparrow$ (Loss$\downarrow$)|Numerical indirect$\uparrow$ (Loss$\downarrow$)|Boolean$\uparrow$ (Loss$\downarrow$)
> -|:-:|:-:|:-:
> ViperGPT|80.6 (19.4$\downarrow$)|31.6 (68.4$\downarrow$)|21.6 (72.4$\downarrow$)
> LLaVA-7B|27.0 (9.9$\downarrow$)|25.0 (15.2$\downarrow$)|58.5 (4.8$\downarrow$)
> LLaVA-13B|24.8 (11.9$\downarrow$)|20.8 (21.2$\downarrow$)|56.3 (4.7$\downarrow$)
> LLaMA-Adapter-7B|30.1 (5.8$\downarrow$)|34.3 (5.6$\downarrow$)|45.8 (14.5$\downarrow$)
>
> 2. ***POPE*** [4] for object hallucinations. POPE evaluates object hallucinations as a binary classification task, and includes 500 images from MSCOCO with 6 questions per sample. As shown in the table below, for different evaluation settings, LLaMA-Adapter with LLaMA-7B attains competitive accuracy compared to other multi-modal LLMs with LLaMA-13B, which indicates our relatively stronger robustness to object hallucination problems.
> Method|Random|Popular|Adversarial
> -|:-:|:-:|:-:
> mPLUG-Owl-7B|53.30|50.63|50.67
> LLaVA-13B|54.43|52.43|50.77
> MM-GPT-7B|50.03|50.00|50.00
> LLaMA-Adapter-7B|75.47|60.43|60.66
>
> #### Reference:
> #### [1] Llama 2: Open Foundation and Fine-Tuned Chat Models. arXiv 2023.
> #### [2] What If the TV Was Off? Examining Counterfactual Reasoning Abilities of Multi-modal Language Models. arXiv 2023.
> #### [3] Making the V in VQA Matter: Elevating the Role of Image Understanding in Visual Question Answering. CVPR 2017.
> #### [4] Evaluating Object Hallucination in Large Vision-Language Models. arXiv 2023.

---

> ### Author Response · Authors · 2023-11-21
> **Response to Reviewer EjGx (Part #2)**
>
> > **Q3: It's better to test the scalability of the method, e.g., using a larger base model or more instruction tuning data**
>
> >
> Thanks for your suggestion! As you suggested, we evaluate the scalability of our LLaMA-Adapter from two aspects: base model size and instruction data size.
>
> 1. **Base Model Size.** In the original paper, we conduct all experiments with LLaMA-7B. To scale up the base model, we further implement LLaMA-Adapter with a larger LLaMA-13B, and compare the results on three benchmarks as follows. On most tasks, LLaMA-13B attains better performance than LLaMA-7B, which demonstrates that our approach can be flexibly adapted to larger LLMs for potentially better multi-modal instruction-following capacity.
> | LLM Size|MME ||  | MMBench |  |  |  |  |  |  | LVLM-eHub |  |  |  |  |
> |:---|:---:|:---:|:---:|:---:|:---:|:---:|:---:|:---:|:---:|:---:|:---:|:---:|:---:|:---:|:---:|
> | LLaMA-Adapter | **All** |P | C | **All** | LR | AR | RR | FP-S | FP-C | CP | **All** | VP | VKA | VR | VC |
> | w LLaMA-7B |1222| 973 | 249 |39.5| 13.1 | 47.4 | 23.0 | 45.0 | 33.2 | 50.6 |0.67| 0.81 | 0.44 | 0.83 | 0.59 |
> | w LLaMA-13B |1299|1021|278|40.1|13.9|47.4|23.7|46.1|34.2|51.2|0.64|0.82|0.39|0.85|0.50|
>
>
> 2. **Instruction Data Size.** By default, we utilize a combination of Alpaca's data (52K) and LLaVA-I (158K) for visual instruction tuning. Here, we progressively add more question-answering data to enlarge the instruction datasets of LLaMA-Adapter: the sampled ***83K VQAv2*** by LLaVA-1.5 [5] and the entire ***204K VQAv2***. As shown in the table below, the increasing instruction tuning data leads to better multi-modal reasoning results on three benchmarks, demonstrating our method’s favorable scalability to data size.
> | Data Size | MME ||  | MMBench |  |  |  |  |  |  | LVLM-eHub |  |  |  |  |
> |:---|:---:|:---:|:---:|:---:|:---:|:---:|:---:|:---:|:---:|:---:|:---:|:---:|:---:|:---:|:---:|
> | LLaMA-Adapter | **All** |P | C | **All** | LR | AR | RR | FP-S | FP-C | CP | **All** | VP | VKA | VR | VC |
> | Alpaca+LLaVA-I |1222| 973 | 249 |39.5| 13.1 | 47.4 | 23.0 | 45.0 | 33.2 | 50.6 |0.6675| 0.81 | 0.44 | 0.83 | 0.59 |
> | +VQAv2 (83K) |1256| 1007 | 249 |43.4| 22.9 | 44.7 | 31.3 | 46.7 | 46.9 | 50.3 |0.6925| 0.84 | 0.42 | 0.88 | 0.63 |
> | +VQAv2 (204K) |1618| 1272 | 346 |60.1| 34.7 | 65.3 | 48.7 | 63.1 | 57.3 | 69.3 |0.7175| 0.86 | 0.44 | 0.92 | 0.65 |
>
>
> #### Reference:
> #### [5] Improved Baselines with Visual Instruction Tuning. arXiv 2023.

---

> ### Author Response · Authors · 2023-11-22
> **Sincere Request for Further Discussions**
>
> Dear Reviewer EjGx,
>
> Thanks again for your great efforts and constructive advice in reviewing this paper! With the discussion period drawing to a close, we expect your feedback and thoughts on our reply. We put a significant effort into our response, with several new experiments and discussions. We sincerely hope you can consider our reply in your assessment. We look forward to hearing from you, and we can further address unclear explanations and remaining concerns if any.
>
> Regards,
>
> Authors

---

### Meta-Review · Area_Chair_aQuE · 2023-12-11

**Metareview:**

The paper introduces a method for efficient fine-tuning of large language/multimodal models, using a learnable gating mechanism to progressively inject instruction knowledge into the model. The reviewers raised concerns mainly about the experimental analysis and comparison with other SOTA models. The author response included new experiments with other models, additional benchmarks, and more recent baselines. Two reviewers recommend acceptance, while one reviewer recommends borderline rejection, arguing the gains over LoRA are marginal in most tasks. While this is a legitimate concern, the AC recommends acceptance, agreeing with the authors that this is not the case for the multimodal setting. As pointed out by reviewer EjGx, the method is simple and yields good results. For the final version, please proofread the paper carefully (e.g., there are mistakes such as section ??)

**Justification For Why Not Higher Score:**

As pointed out by reviewer WSmj, the method is incremental in nature and offers marginal gains over LoRA (for language tasks). These are legitimate concerns

**Justification For Why Not Lower Score:**

The method is simple to implement and offers good results, especially in the multimodal setting.

---

### Decision · Program_Chairs · 2024-01-16

Accept (poster)